# Atlantic Masters: Three Early Modern Afro-Brazilian Artists

Miguel A. Valerio

Romance Languages and Literatures, Washington University in St. Louis, St. Louis, MO 63130, USA; m.a.valerio@wustl.edu

**Abstract:** Brazil received the largest number of Africans enslaved into the Americas: nearly five million by some estimates. Thus, Brazil became the world's largest slavocracy. But slavery was not the only experience available to Africans and Brazilians of African descent in slavery-era Brazil. Numerically, Afro-Brazilians dominated the arts in colonial Brazil. However, very few of those artists and artisans, many of whom were enslaved, are known by name today. Free Afro-Brazilian artists, such as Aleijadinho, Mestre Valentim, and Teófilo de Jesus, on the other hand, fared far better. In this article, I turn to these three mixed-race artists' works and what little is known of their lives, not only as exemplary Afro-Brazilian artists but also as some of the most important artists of Brazil's late colonial period, where they had the greatest impact on the artistic developments in their home regions. These artists' careers thus illustrate how artists of African descent contributed to and defined urban and sacred spaces in the early modern Atlantic. This is therefore an invitation to look at Afrodescendants' role in early modern art beyond the anonymity of slavery and static representation.

**Keywords:** colonial Brazil; public works; sacred art; sculpture; urban planning; painting; Antônio Francisco Lisboa; Aleijadinho; Valentim de Fonseca e Silva; Mestre Valentim; Jose Teófilo de Jesus





## 1. Introduction

Brazil received the largest number of Africans enslaved into the Americas: nearly five million by some estimates (Slave Voyages 2021; Klein and Luna 2010). Thus, Brazil became the world's largest slavocracy, especially for most of the eighteenth and nineteenth centuries, as demand for slave labor was propelled by the rise of mining in the early eighteenth century and the coffee industry starting in the late eighteenth century. Brazil was also the last American nation to abolish slavery in 1888. But slavery was not the only experience available to Africans and Brazilians of African descent during this period. Numerically, Afro-Brazilians (whether born in Brazil or elsewhere) dominated the arts (painting, sculpture, architecture, and music) and the trades (e.g., metalwork, cobblery, and barbering) in colonial Brazil (1500–1822) (Araújo [1988] 2010, vol. 1, pp. 25–182; Furtado 2015; Campos 2003; Querino [1911] 2018; Valerio 2021a).[1] However, very few of those artists and artisans, many of whom were enslaved, are known by name today. This is because, as enslaved artists and artisans, these Afro-Brazilians' works often bear the names of their enslavers (see Santiago 2008; Furtado 2015; Campos 2005, pp. 199–211). In other words, these enslaved artists and artisans did not receive individual commissions, but rather worked for enslavers, such as colonial Brazil's most celebrated Luso-Brazilian, or White artists, such as Manoel da Costa Ataíde (1762–1830) in Minas Gerais (Frota 1982; Campos 2005; Santiago 2008) and José Joaquim da Rocha (ca. 1737–1807) in Bahia (Querino [1911] 2018, pp. 89–93; Ott 1982, pp. 10–74).

Free Afro-Brazilian artists, some of whom also enslaved other Black artists, such as the mixed-race Antônio Francisco Lisboa (Ouro Preto, ca. 1730–1814; active in Minas Gerais, 1760s–1814), more commonly known as Aleijadinho, Valentim de Fonseca e Silva (Serro, ca. 1745—Rio de Janeiro, 1813; active in Rio, 1770s–1813), better known as Mestre Valentim, and José Teófilo de Jesus (Salvador, ca. 1758–1847; active in Salvador, 1790s–1847), on the other hand, fared far better. Unlike enslaved Black Brazilians, these mixed-race free

artists received individual commissions. Aleijadinho and Valentim were, in fact, the most sought-after artists in their lifetime.

In this article, I turn to Aleijadinho, Valentim, and Jesus' works and what little is known about their lives, not only as exemplary Afro-Brazilian artists but also as some of the most important artists of Brazil's late-colonial period, where they had the greatest impact on the artistic developments in their home regions. These artists' careers thus illustrate how artists of African descent contributed to and defined urban and sacred spaces in the early modern Atlantic. This is therefore an invitation to look at Afrodescendants' role in early modern art beyond the anonymity of slavery and passive representation. This article thus contributes to the scholarship and efforts that are bringing greater institutional recognition to Black artists of the Atlantic during the age of slavery (1500–1888) (see, e.g., Pullins and Valdés 2023; Spinozzi 2022) and is a reminder that many of colonial Latin America's most celebrated artists, today and in their lifetimes (such as Aleijadinho in Brazil and Juan Correa (Mexico City, 1646–1716; active Mexico City; see, e.g., Vargaslugo 2017) in Mexico) were of African descent.[2] This in turn should compel us to reconsider how art provided interstices in the region's growing racial regimes (see, e.g., Pessoa 2013; Viana 2007; Martínez 2008; Nemser 2017), cracks that allowed some of these artists to enter the halls of power, as Valentim did, as artists rather than as enslaved servants or defendants.

Aleijadinho, Valentim, and Jesus each lived, worked, and left their mark on the three most prosperous and powerful regions of colonial Brazil. Aleijadinho lived and worked in Minas Gerais, where the mining of gold and gems ushered in a Portuguese Golden Age (see Boxer 1962) that reached far and wide and filled the region with ostentatious sanctuaries that sought Aleijadinho's imprint. Valentim was based in Rio de Janeiro, which became the capital of colonial Brazil in 1763 and of the Portuguese Empire in 1810, shortly before his death. Finally, Jesus resided and worked in Salvador, which had previously been the capital of colonial Brazil and, as such, was full of stunning churches always looking for artists to update their interiors to the latest style. Although Valentim is best known for his public work, all three artists benefitted from the unique situation in Brazil of powerful third orders, or lay Catholic associations mostly made up of the elite, that competed among themselves to build the most ostentatious churches (Boschi 1986; Bazin 1956–1958, [1956–1958] 1983; Bailey 2014, pp. 177–236). These orders did not admit anyone with Muslim, Black, or Jewish ancestry (de Salles 2007), which means that these artists would have not been admitted into their ranks.

While scholars have looked for how Afro-Brazilian artists imprinted their ethnoracial identity on their works, no such traces are to be found in Aleijadinho, Valentim, and Jesus' works (see Zimmerman in this issue; Conduru and Ribeiro da Silva Bevilaqua 2022; Tribe 1996; Sullivan 2006). These artists instead worked in the rococo style of the late Baroque. Brazilian rococo is unique at both the national and regional levels. At the national level, it is distinct from European rococo. Taking its cue from German rococo, which arrived via Portugal, the Brazilian variant is less ornate (Bailey 2014, pp. 177–236; de Oliveira 2003, 2014; Oliveira and Honor 2019; de Toledo 1983, 2015). At the reginal level, northeastern rococo (in Bahia and Pernambuco) has more sharp angles than the rounder mineiro rococo (in Minas Gerais) (Bazin 1956–1958, [1956–1958] 1983). Aleijadinho exceled in religious sculpture in stone; Valentim led in public works; and Jesus led in sacred painting. Aleijadinho was the first artist to create large-scale sculptures in the ubiquitous grey *pedra-sabão* (soapstone) of Minas Gerais, and Valentim was the first to cast large-scale bronze sculptures in Brazil.

## 2. Aleijadinho, Master Sculptor

Twenty-eight years after Aleijadinho (Figure 1) died, Rodrigo José Ferreira Bretas (1815–66), a professor and provincial representative, published the first biography of the artist, *Traços biográficos* (Bretas [1858] 2013). Bretas' biography was purportedly based on archival sources and interviews with residents of Ouro Preto, including Aleijadinho's daughter-in-law, Joana, who supposedly cared for the artist in his last days. But very little of what Bretas wrote can be verified. Bretas instead created the myth of Aleijadinho, in whose

mist the real Aleijadinho languishes still. For although Aleijadinho is the most studied artist of colonial Brazil and enough of a popular figure to be a household name, very little about his life is known, albeit this has not stopped the biographers (e.g., Freudenfeld [1943] 1961; Jorge [1949] 2006; G.B. de Carvalho [1956] 2011) from presenting conjectures as facts (see Zimmerman in this issue; de Grammont 2008).

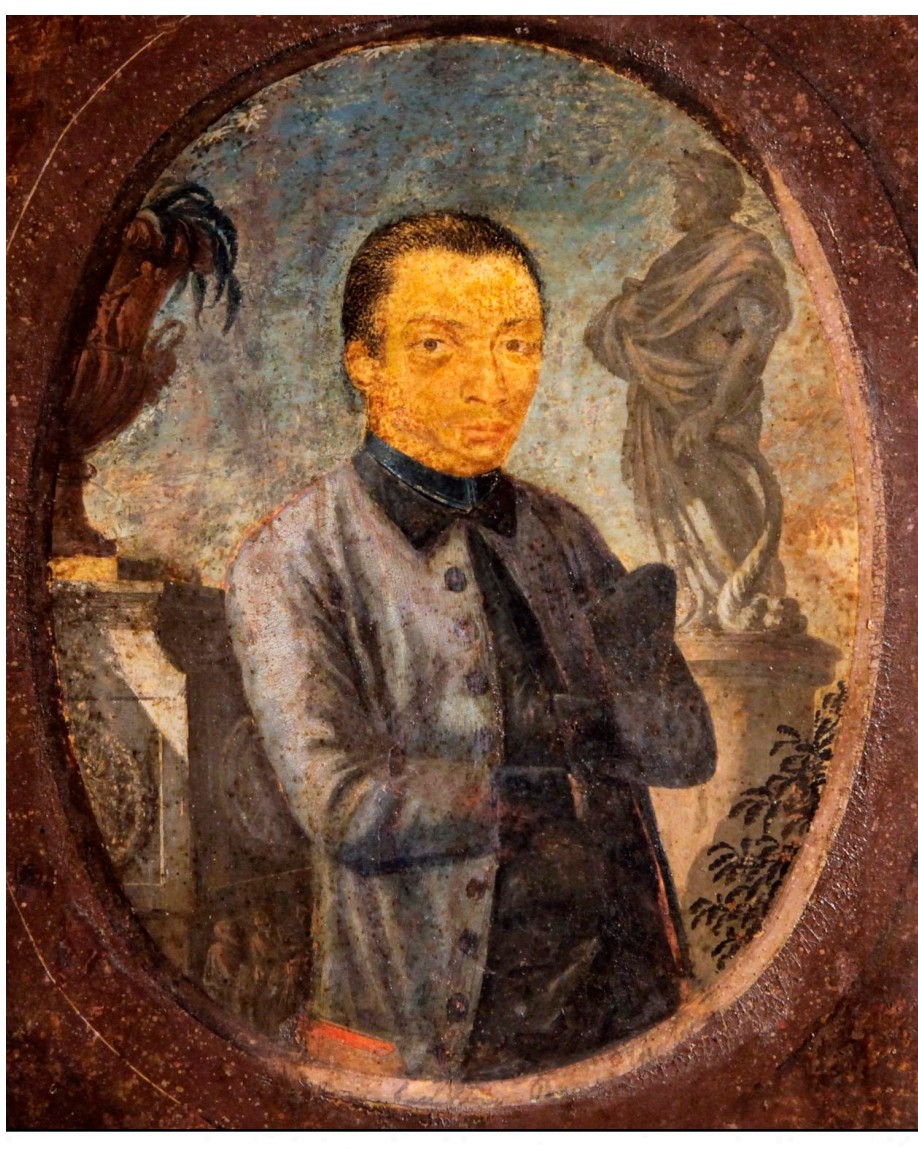

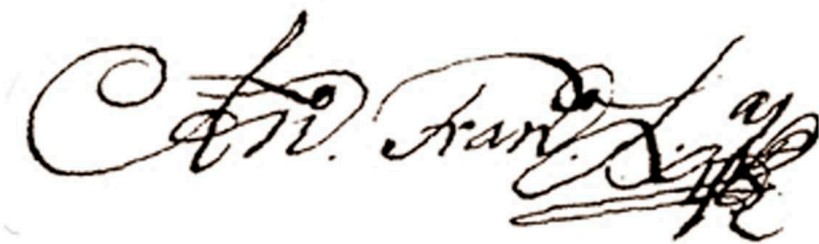

**Figure 1.** Euclásio Ventura, supposed posthumous portrait (19th century) and signature of Aleijadinho. Courtesy of Wiki Commons.

We know that Aleijadinho was baptized on 29 August 1730 (in Jorge [1949] 2006, p. 3). On that occasion, his father's name was given as Manoel Francisco da Costa, who is believed to have been a Portuguese architect active in Ouro Preto. His mother, Isabel, was one of his father's Black slaves. Aleijadinho was thus born a slave, as legal status was inherited from the mother. But his father freed him at his baptism, which was a common practice in urban areas and mining regions for the offspring of slave owners and enslaved women. We do not know anything about his childhood or training. Who did he learn sculpting from? Among the celebrated woodcarvers (*entalhadores*) active in Ouro Preto at the time was Francisco Xavier de Brito (died 1751). Brito's most important work was the high altar of the Basilica do Pilar, the town's parish church, whose consecration in 1734 was immortalized in a very well-known text in Brazil among specialists (Machado 1734). Another important woodcarver was João Gomes Batista (17–18th centuries; dates unknown), who worked in the Carmo church, where Aleijadinho performed his last commission. Moreover, Aleijadinho's uncle, Antonio Francisco Pombal, was also an *entalhador*. Unfortunately, none of the stone sculptors active in Ouro Preto at the time are known by name.

But the greatest mystery and appeal about Aleijadinho for many scholars and the popular imagination is the unknown illness that earned him the epithet *o aleijadinho*, or the cripple. Like everything else about Aleijadinho, much has been conjectured about the nature of his malady, but very little about it is known. Some scholars have proffered that it was leprosy, that he lost his fingers and the use of his legs (Freudenfeld [1943] 1961; Jorge [1949] 2006; de Barbosa [1984] 1988; de Oliveira 1985). The only contemporary source for Aleijadinho's illness is the German geologist Wilhelm Ludwig von Eschwege (Hessen, 1777–1855), who saw the artist in Congonhas in 1811 and wrote in his diary (published in 1818) that "the preeminent sculptor, who distinguished himself here, is a cripple man with lame hands, he has the chisel strapped onto himself and therewith carries out the most artistic works" (von Eschwege 1818, p. 132).

Though the nature of Aleijadinho's illness is unknown, he makes for an important case study about the intersection of art and disability, whose scholarship until now has mostly focused on the representation of disability in the arts (e.g., Siebers 2010; Cooley and Fox 2022; Millett-Gallant and Howie 2017; Watson and Hiles 2022). Aleijadinho offers us the opportunity to consider the artist—often one of society's ablest persons—as a disabled individual. Where it not for the epithet with which Aleijadinho has passed into posterity, that his illness is unknown may mean that his illness did not define him.

Aleijadinho worked exclusively in Minas Gerais but did work in almost every major town in the region: Ouro Preto, Congonhas, Tiradentes, São João del-Rei, and Sabará (see Figure 2). In Ouro Preto, his major work was in the São Francisco (St. Francis) church, which was his first major commission (Figure 3). He did the work on the façade, the pulpits, and parts of the main and side altars (Trinidade [1951] 1958, pp. 133–44). On the façade, Aleijadinho fashioned a sumptuous frame for the door. The frame is crowned with a representation of the Immaculate Conception, the patroness of the Portuguese Empire since 1640 (Figure 4). Above the virgin, Aleijadinho placed a medallion of St. Francis in prayer (Figure 5). These beautiful carvings greeted believers as they approached the church, Ouro Preto's most visited, and called them to prayer. Aleijadinho's work for the São Francisco church made him famous throughout Minas Gerais and beyond, and the most in-demand sculptor in the region for the rest of his life.

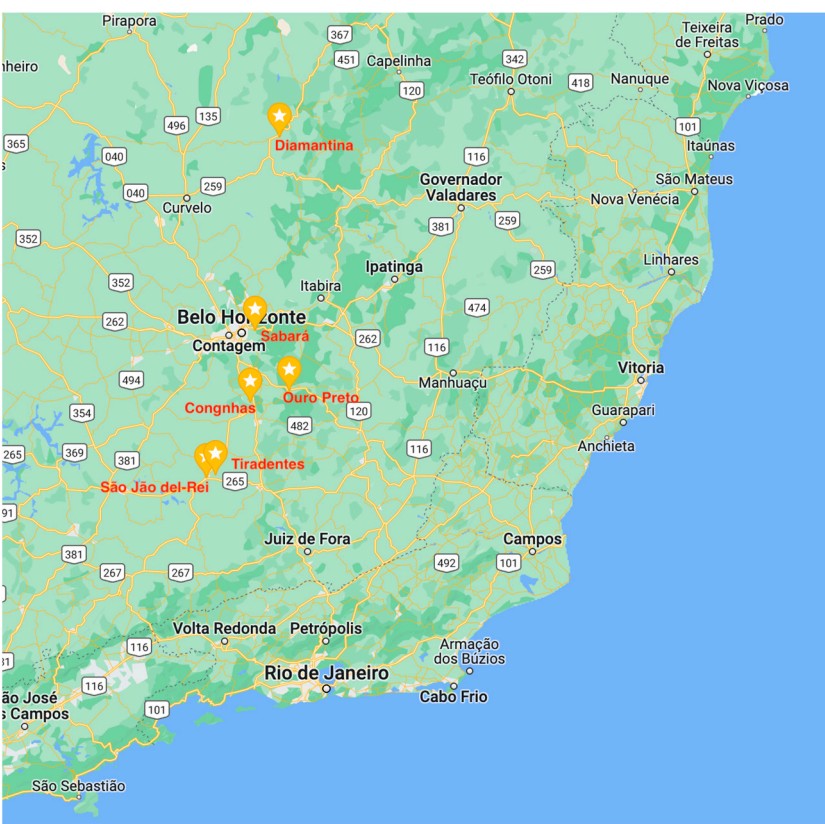

**Figure 2.** Towns where Aleijadinho did work. Google Maps.

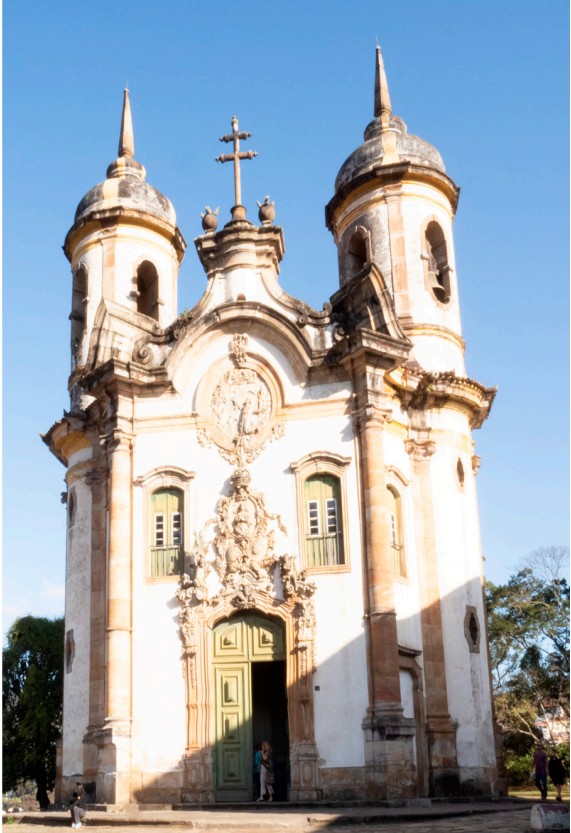

**Figure 3.** Façade of São Francisco, relief work by Aleijadinho, 18th century. Photo by the author. June 2022.

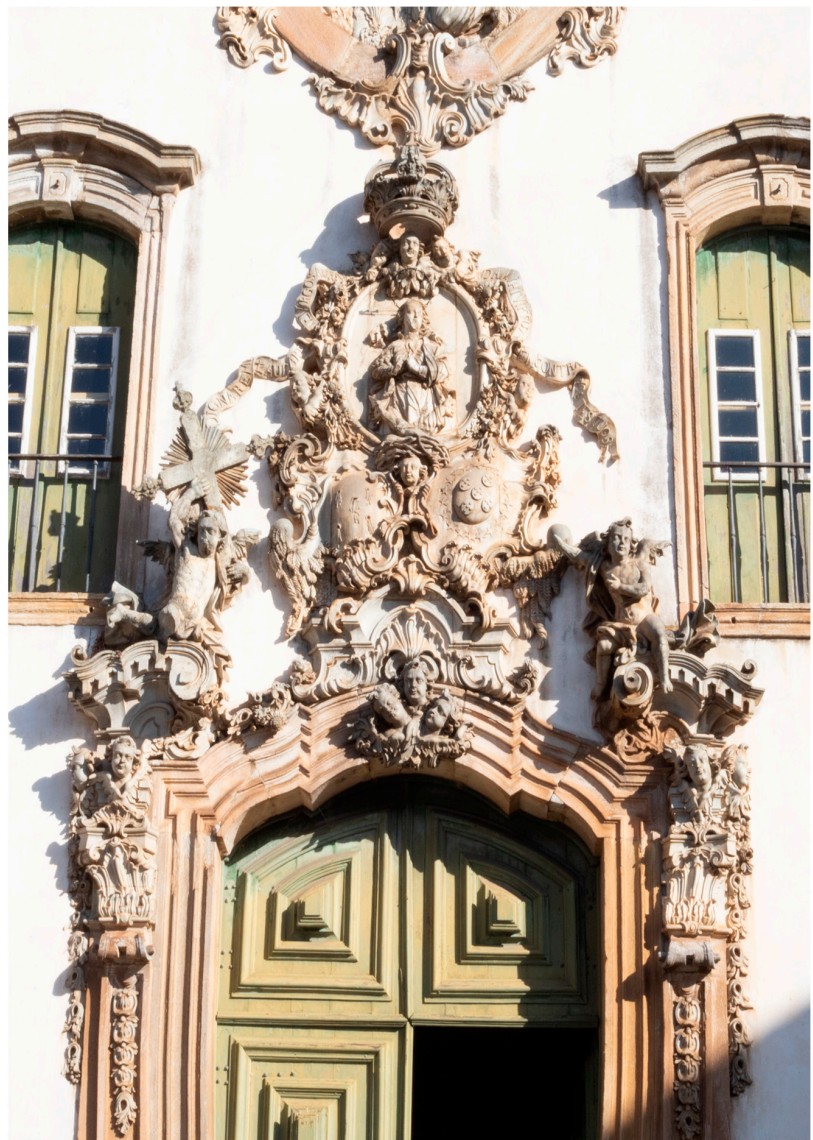

**Figure 4.** Virgin above the door, detail of Aleijadinho's relief on façade of São Francisco, 18th century. Photo by the author, June 2022.

After surviving an illness, a landed gentleman, Feliciano Mendes, promised to honor an image of Jesus under the title Bom Jesus de Matosinhos (Good Jesus of the Little Forest) and built a chapel in Congonhas, about thirty miles from Ouro Preto. This chapel became a place of pilgrimage, as the image was held to be miraculous, which in turn generated a great deal of revenue for the sanctuary. Thus, in the 1790s, the lay Catholic third order (*ordem terceira*) that ran it chose to enlarge it. The *ermitão*, or lead brother, overseeing the work thus called the region's most prominent artists, Ataíde and Aleijadinho, to work on the sanctuary (de Oliveira 1985, pp. 15–18; 2006; França 2015).

In Congonhas, Aleijadinho sculpted ten Old Testament prophets in *pedra-sabão* for the entryway into the new church and sixty-six figures in cedar for the passos (Way of the Cross) housed in five structures downhill from the sanctuary (Figures 6–8). Both the prophets and the passion figures are monumental, but statues of such a great size had never been attempted in *pedra-sabão*. Aleijadinho worked with a team of pupils, including the artists whom he enslaved, to complete the project (de Oliveira 1985, pp. 15–18; 2006; França 2015). The prophets are among the best-known statues in Brazil. Aleijadinho sculpted each prophet with a scroll with a bible quote (Figures 9 and 10). No other sculptor contributed as much as Aleijadinho to Minas Gerais' sacred spaces (de Oliveira et al. 2008).

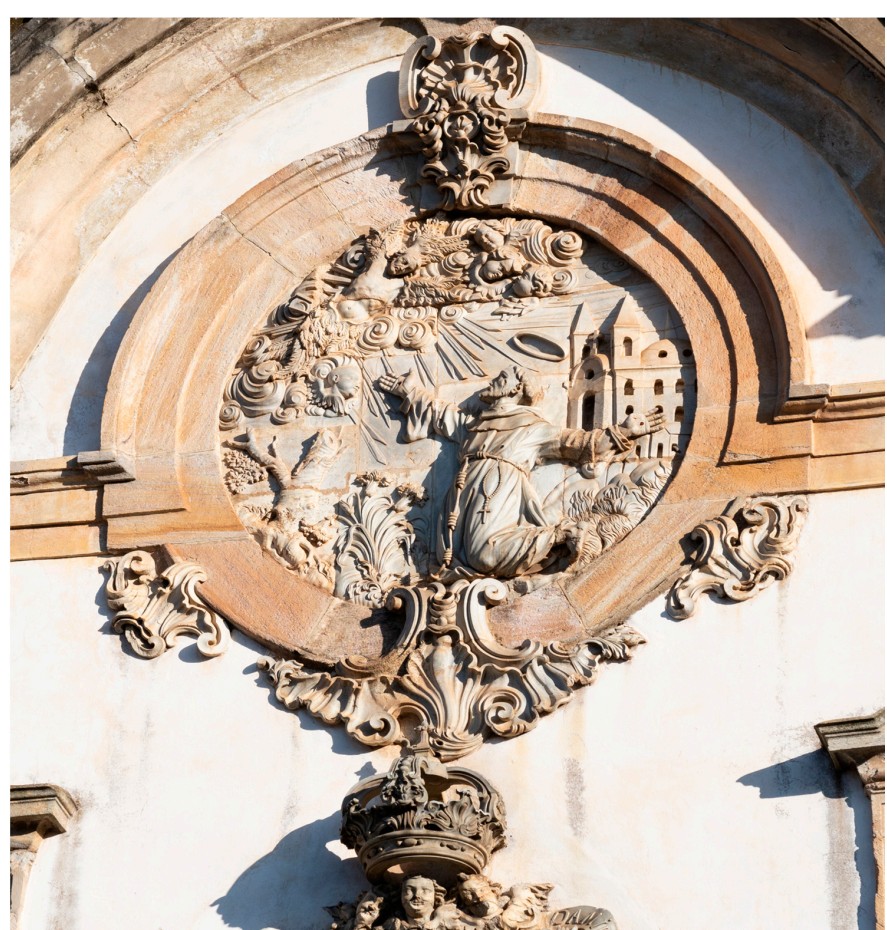

**Figure 5.** St. Francis medallion, detail of Aleijadinho's relief on façade of São Francisco, 18th century. Photo by the author, June 2022.

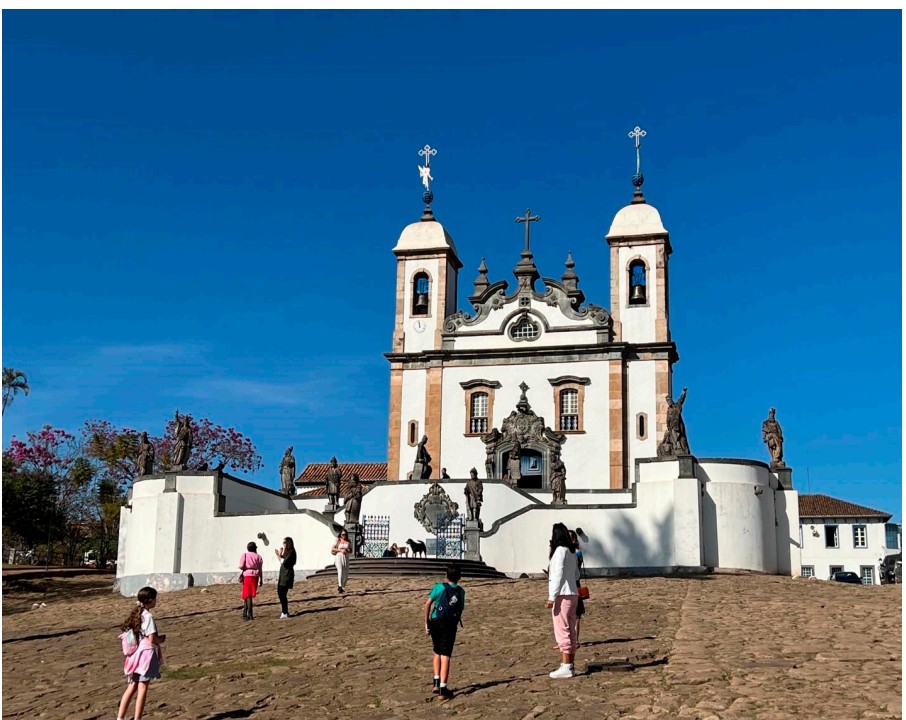

**Figure 6.** Basilica of Bom Jesus de Matosinhos, 1795–1805. Photo by the author, June 2022.

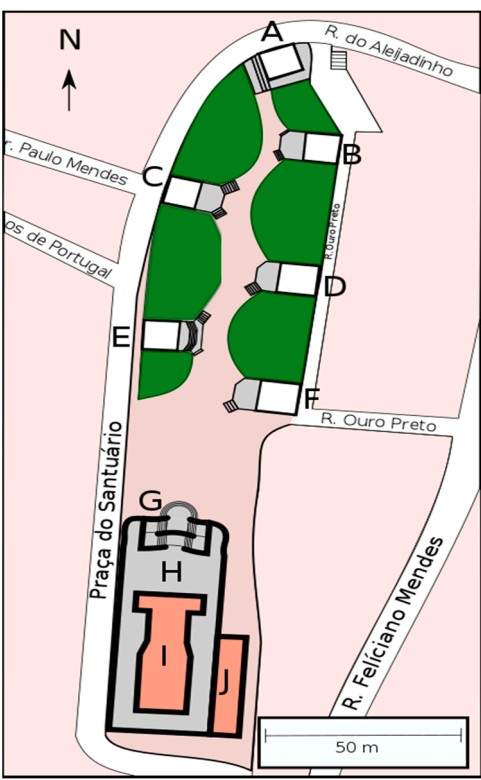

**Figure 7.** Layout of the sanctuary of Bom Jesus de Matosinhos. A–F are the buildings with the stations of the cross. G are the prophets. H is open space in front of church. I is the church. J is ex-voto room or "Room of Miracles." Courtesy of Wiki Commons.

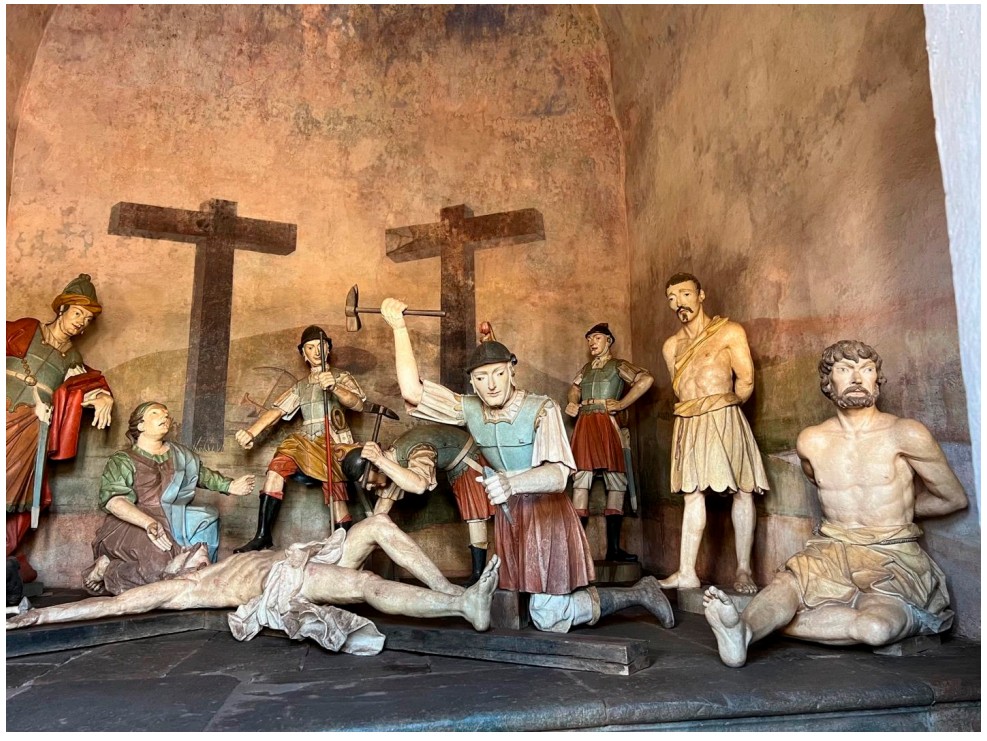

**Figure 8.** Aleijadinho, The Crucifixion, 1795–1805. Photo by the author, June 2022.

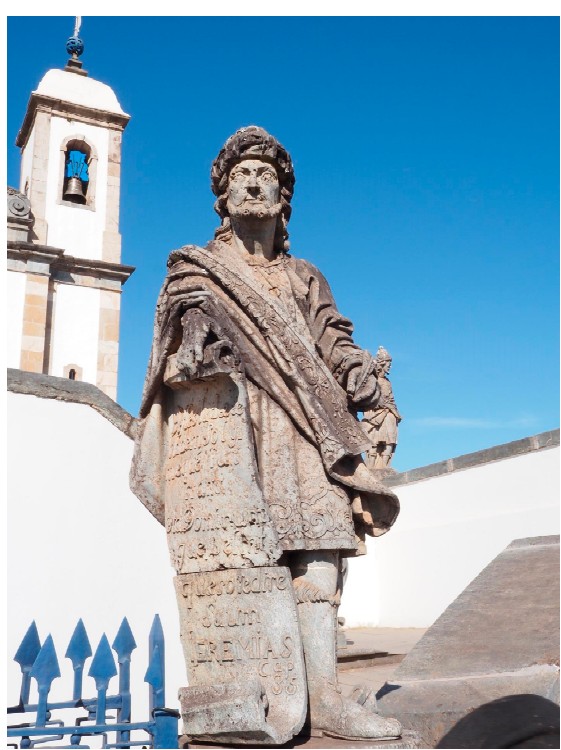

**Figure 9.** Aleijadinho, Jeremiah, 1795–1805. Photo by the author, June 2022.

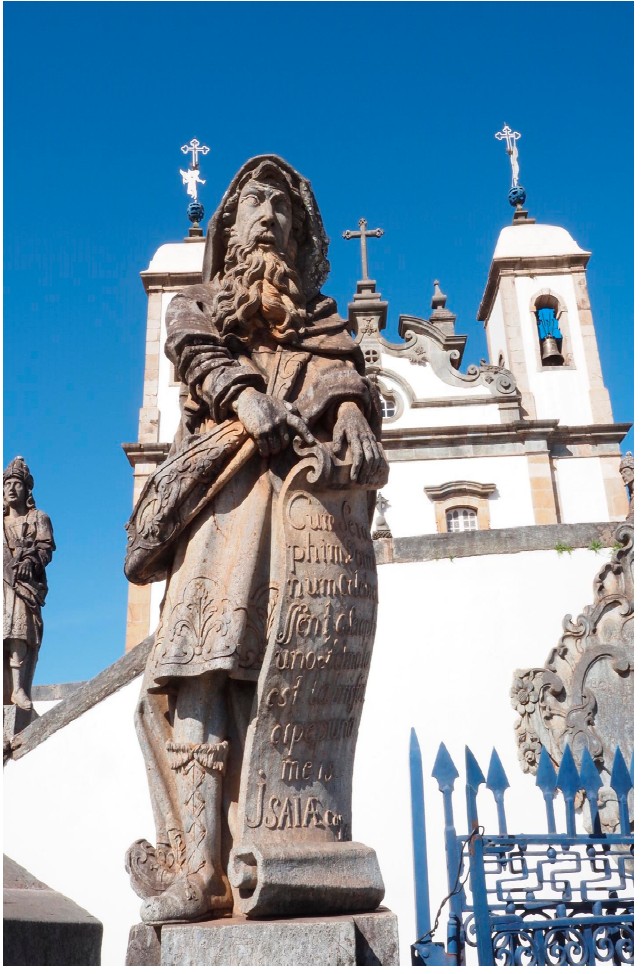

**Figure 10.** Aleijadinho, Isaiah, 1795–1805. Photo by the author, June 2022.

The influence of European prints on Aleijadinho's work has long been recognized (e.g., Bazin 1963; de Oliveira 1979, 2003). The French art historian Germain Bazin (1963), for example, found many European parallels in Aleijadinho's passion figures and prophets. Bazin compared Aleijadinho's Jesus to Pompeo Batoni's *Sacred Heart* (pp. 223–54). The resemblance is indeed striking. Bazin likewise found a set of Florentine prints that bore some resemblance to Aleijadinho's prophets (pp. 263–75).

### 3. Valentim, Master of Public Works

Despite his prominence during his lifetime, Valentim is far less studied than Aleijadinho. Like Aleijadinho, Valentim was born in Minas Gerais, in Serro—near today's state capital, Belo Horizonte. There is no evidence that the two artists ever met, however. Valentim is the only one of these artists for whom we have an autobiographical document: his last will and testament, notarized on 2 February 1813, five days before his death. He calls himself the illegitimate son (*filho natural*) of Manoel da Fonseca e Silva, who is believed to have been a diamond inspector, and Amatilde da Fonseca, his father's slave (da Fonseca e Silva [1813] 2008, p. 143; Cavalcanti 2004, pp. 310–13). Valentim was thus also born a slave. We do not know anything else about the artist until he shows up in Rio in the 1770s.[3] Like Aleijadinho's, his training is a mystery. But he was trained to work with wood, stone, and metal (e.g., Figure 11). He thus left the most varied legacy of these three artists. In his will, Valentim names an illegitimate daughter, Joana, whom he had with a certain Josefa Maria da Conceição, whom he did not marry, though there was no legal "impediment" (da Fonseca e Silva [1813] 2008, pp. 142–43), which means that Josefa was also of African descent. Joana, however, was not raised by her mother, but rather by a certain Teodora Maria dos Santos (da Fonseca e Silva [1813] 2008, p. 142). At the time of his death, Valentim stated that he had been in Rio for "more than forty years" (da Fonseca e Silva [1813] 2008, p. 143). He also freed his slave, Antônio Mina, upon his death (da Fonseca e Silva [1813] 2008, p. 143).

In Rio, Valentim set up shop on the Rua do Sabão (part of present-day Avenida President Vargas—today's downtown Rio's widest and busiest avenue). He joined the Catholic Brotherhood of Our Lady of the Rosary and Saint Benedict of Palermo of the Blacks, whose church (on present-day Rua Uruguaiana and Rua do Rosário) had been made the city's cathedral against the brotherhood's wishes in 1737 (de Carvalho Soares 2011, pp. 113–25). Valentim may have contributed to the church, especially its rococo interior, which burned down in 1967 (Figures 12 and 13). When he died in 1813, the master was buried in this church.[4]

Valentim's first commissions in Rio were the interior of the church of the Third Carmelite Order and two silver lamps for the church of the city's Benedictine monastery, São Bento (Figure 14). As his fame grew, Valentim became the favorite artist of viceroy Luís de Vasconcelos (r. 1778–90).[5] When a fire destroyed the Recolhimento do Parto (Safehouse of Our Lady of Birth), which housed destitute women and was near Valentim's residence, the viceroy, the recolhimento's patron, hired the master to design its rebuilding.[6] This event left us the only portrait of an Afro-Brazilian colonial artist, by the Italian-born João Francisco Muzzi, in which Valentim is handing the design to the viceroy (Figure 15). Through their partnership, Vasconcelos and Valentim changed Rio. As the viceroy wanted to rid the city of its fetid lagoons and ponds, he entrusted Valentim to execute his urban reform projects, which became the artist's most famous works.

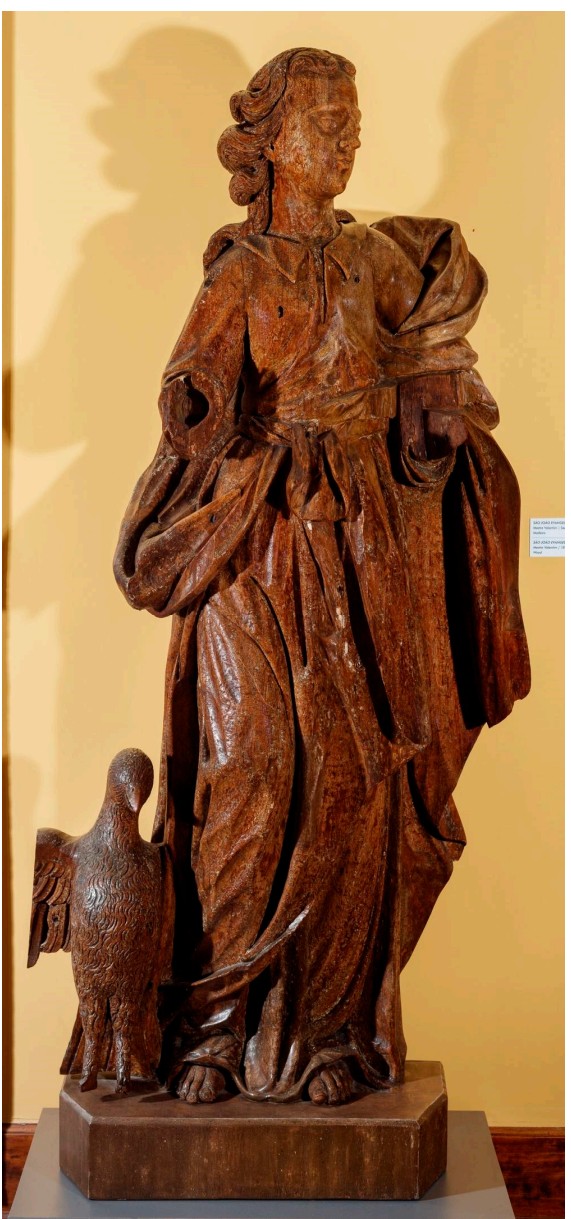

**Figure 11.** Valentim da Fonseca e Silva, St. John the Evangelist, ca. 1800, Museu Histórico Nacional. Courtesy of the Museu Histórico Nacional.

The first thing that a late eighteenth- or nineteenth-century traveler arriving by sea in Rio would see was Valentim's Pyramid fountain (1789; Figures 16 and 17). The traveler would then step into the Largo do Carmo (Largo do Paço in the nineteenth century, Praça XV today), which Valentim had also reformed. This space was also where enslaved Africans arrived and were sold before the slave port was moved to Valongo (a wharf on the north side of the city center, north of Praça XV) in the nineteenth century for health reasons. Valentim built three other fountains for Rio: the Lizard (*Lagarto*) (Passeio Público, ca. 1780), Marrecas (1785, destroyed in 1896 (A.M.F.M. de Carvalho [1999] 2003, p. 35); Figure 18), and Saracuras (1795); (Figure 19) (Marianno 1943; Ribeiro 2000). These were not merely decorative fountains, as most were the sources of potable water for the citizenry, as we see in Figure 18. But Valentim made them aesthetically pleasing by adding decorative elements, especially bronze sculptures of animals, to all of them (e.g., Figure 20). The sculpted nature Valentim placed in his fountains (lizard, turtle, cranes, caimans, etc.) is decidedly Brazilian.

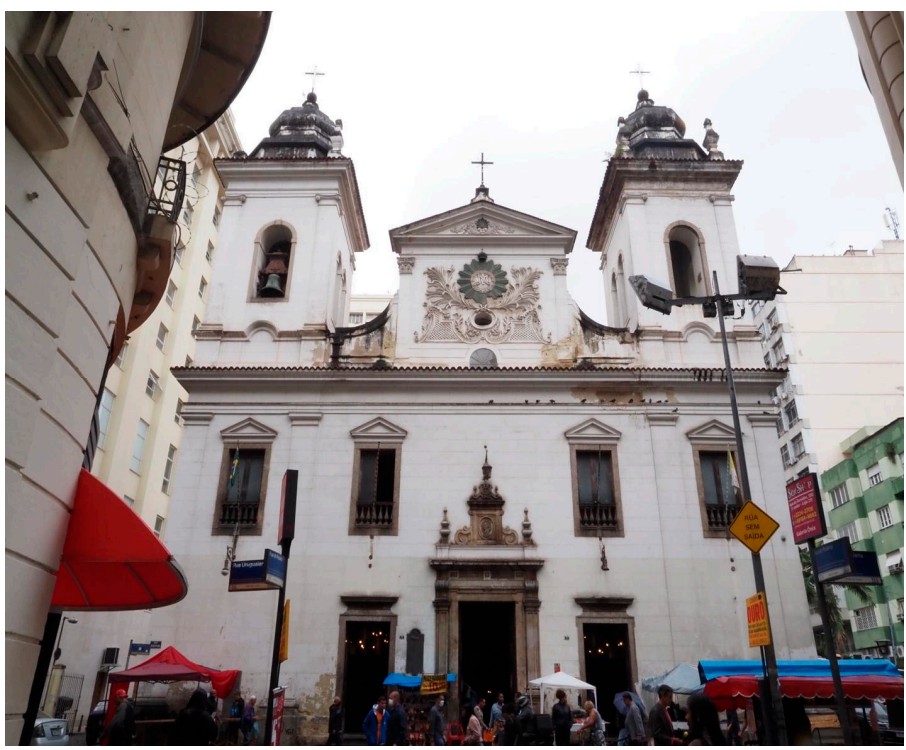

**Figure 12.** Unknown architects, Rosario church, 18th century, Rio de Janeiro. Photo by the author, January 2023.

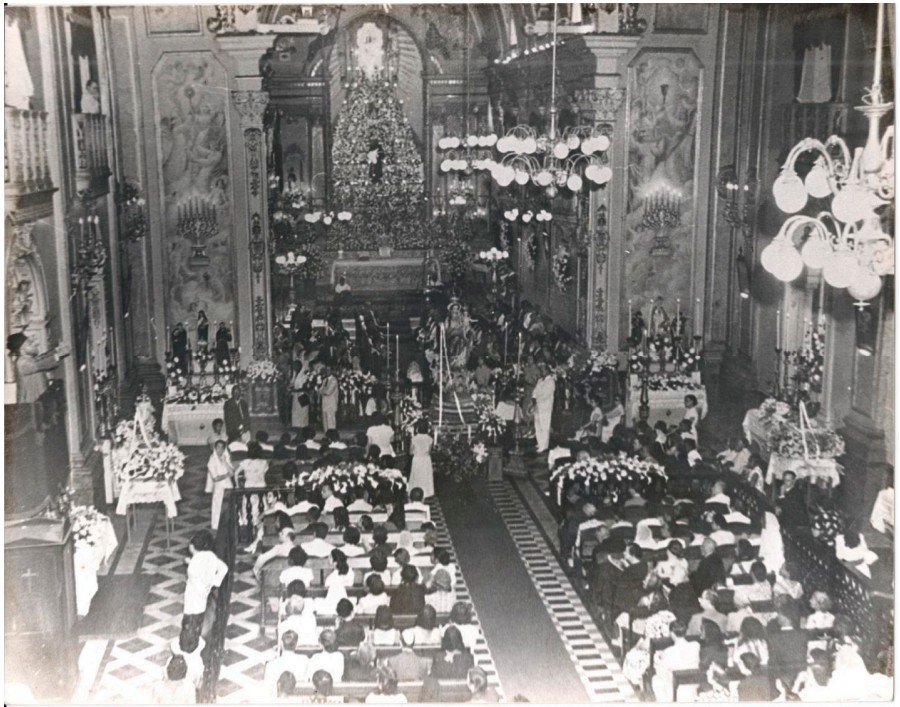

**Figure 13.** Unknown artists, interior of Rosario church before the 1967 fire. Photo courtesy of the Rosario Church.

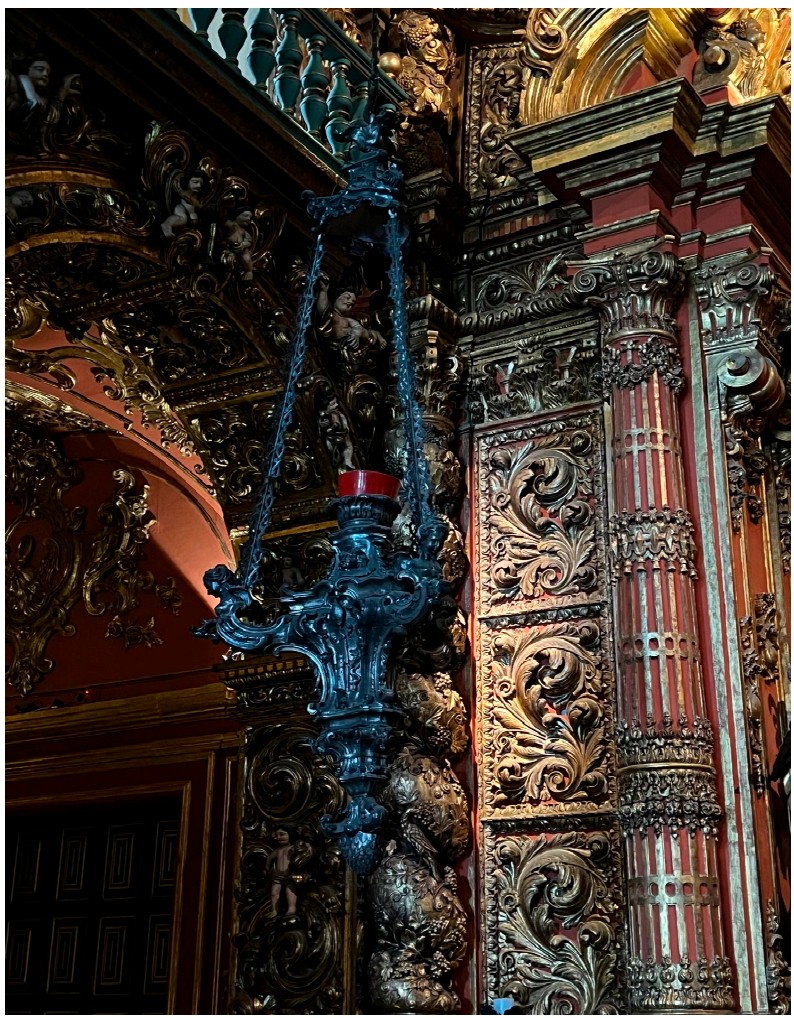

**Figure 14.** Valentim da Fonseca e Silva, lamp for the São Bento monastery church, 18th century. Photo by the author, January 2023.

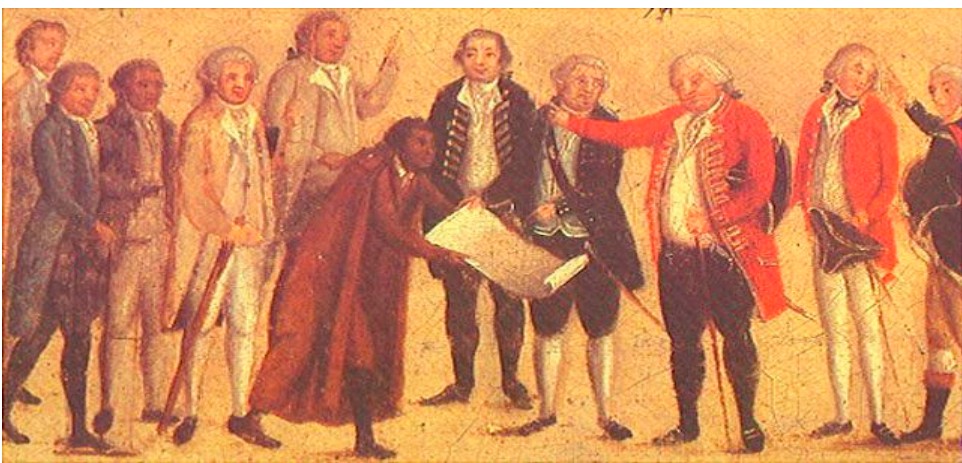

**Figure 15.** João Francisco Muzzi, Valentim hands plans for the new recolhimento to the viceroy, *Joyfull and Quick Rebuilding of the Old Nossa Senhora do Parto Recolhimento from August 25 to December 8*, detail, 1789, Museu Castro Maya. Courtesy of the Museu Castro Maya.

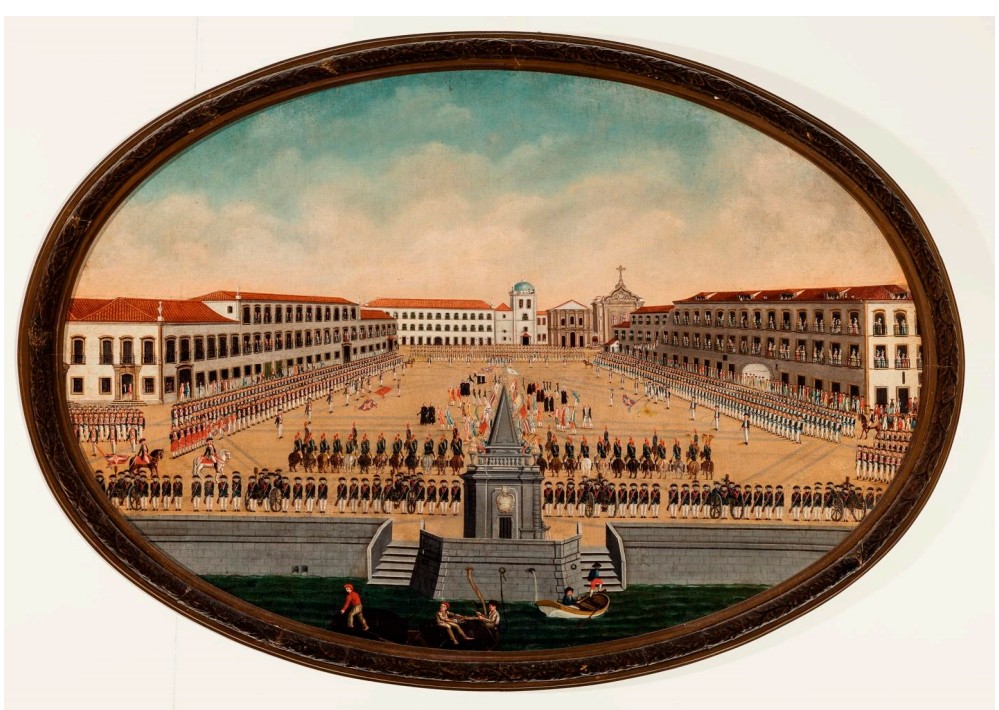

**Figure 16.** Attributed to Leandro Joaquim, *Military Review in the Largo do Paço*, c. 1780. Courtesy of the Museu Histórico Nacional.

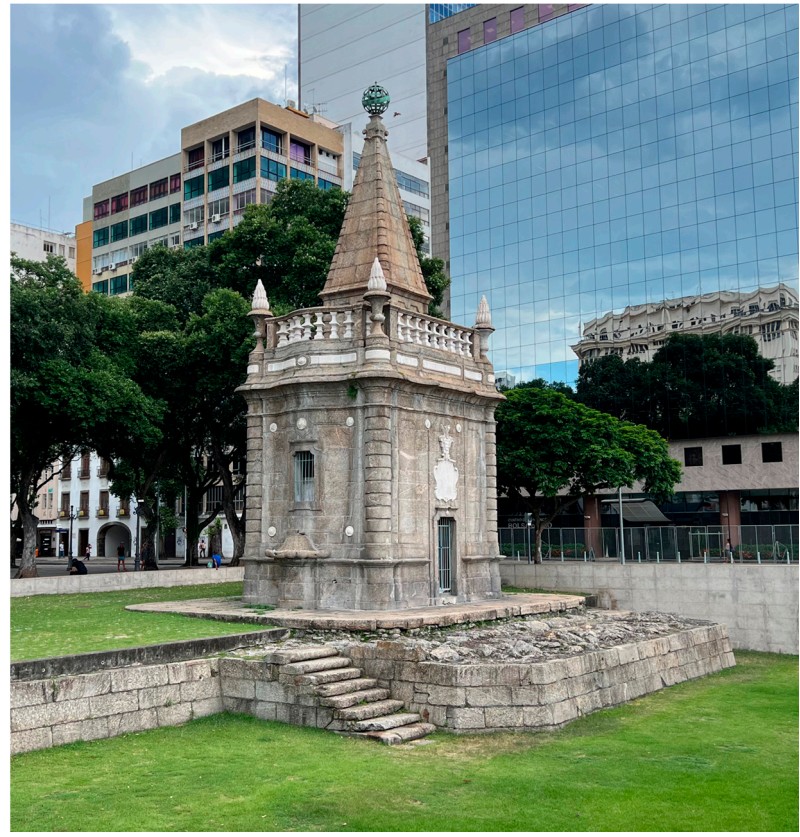

**Figure 17.** Pyramid fountain today in Praça XV, Rio de Janeiro. Photo by the author, January 2023.

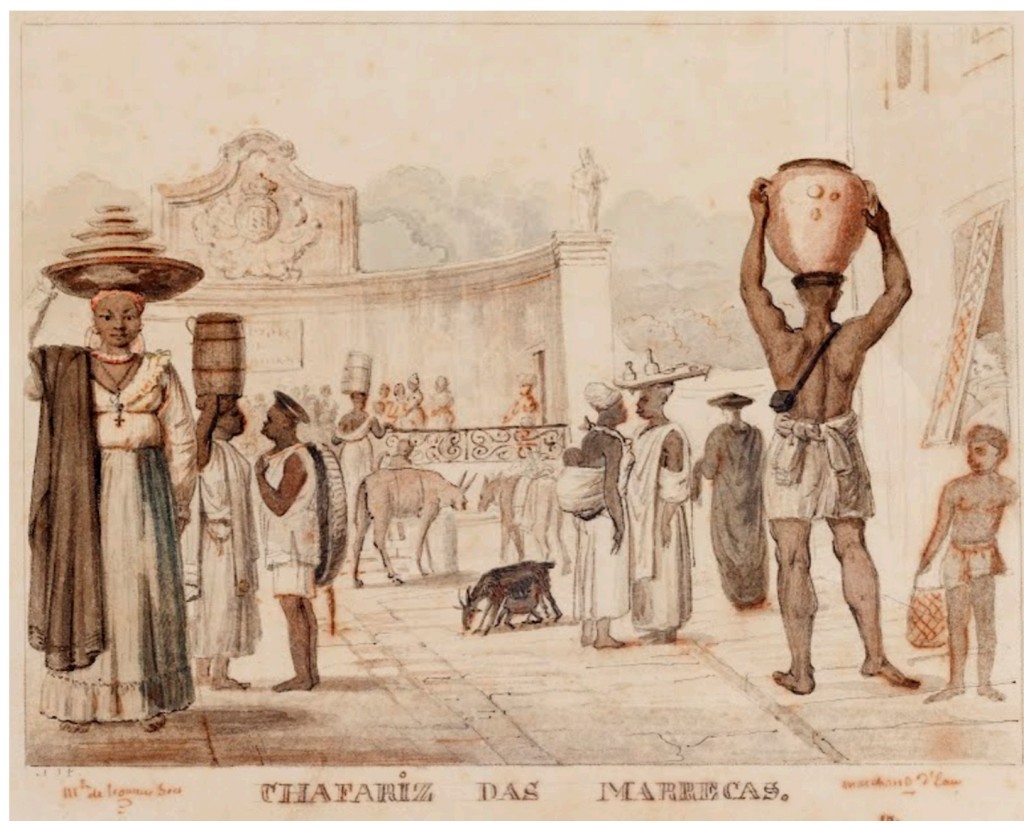

**Figure 18.** Armand Julien Palière (French, 1784–1862), Marrecas fountain, 1830/39. Courtesy of the Museu Histórico Nacional.

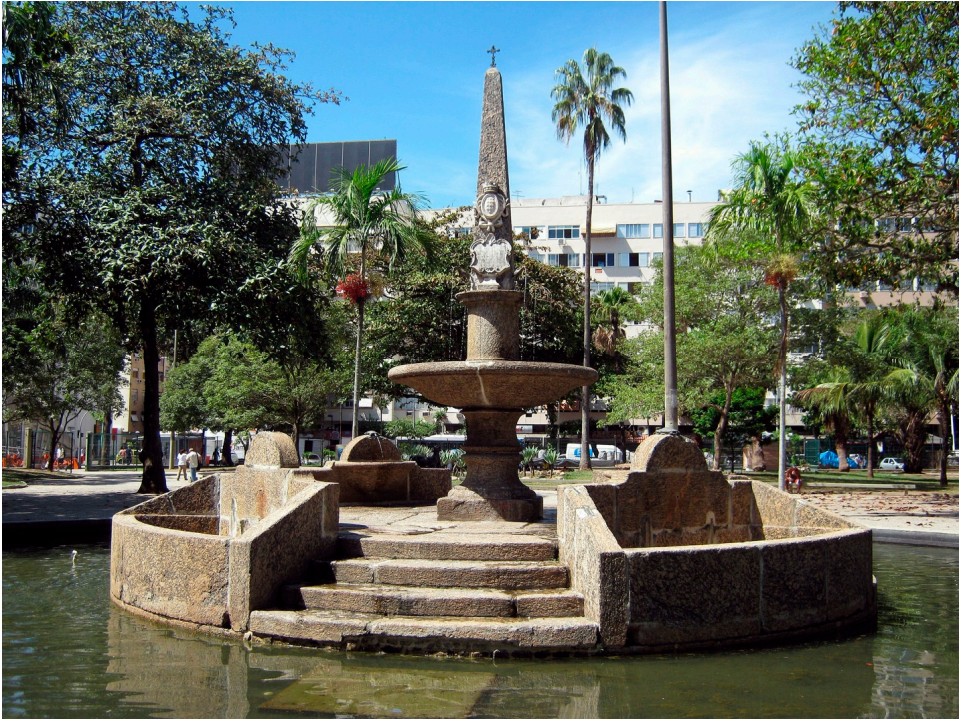

**Figure 19.** Saracuras fountain today. Courtesy of Wiki Commons.

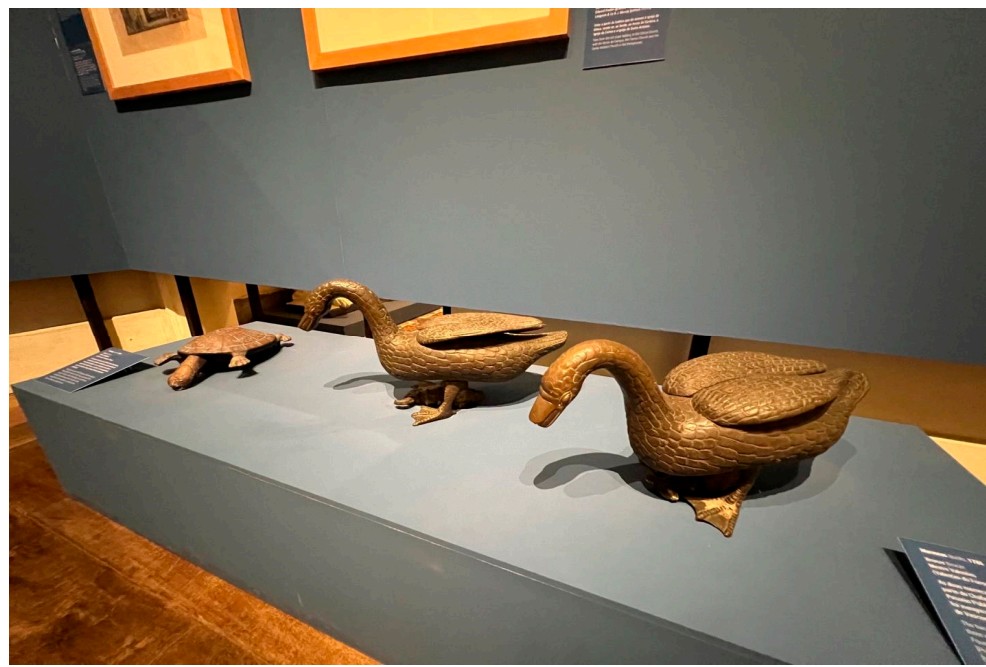

**Figure 20.** Animals from the Pyramid fountain, Museu Histórico da Cidade, Rio de Janeiro. Photo by the author, January 2023.

Valentim's largest project would be the Passeio Público (Public Promenade, 1779–83), which replaced a lagoon that was a public health hazard, as waste from the city's hospital was dumped there (Cavalcanti 2004, p. 43); (Figure 21). The Passeio was the first public park in the Americas designed for leisure. In its original form, it was a promenade at the water's edge south from Largo do Carmo, as we can see in the center of the German diplomat Karl Wilhelm von Theremin's (Berlin 1784–1852) depiction of the entrance (Figures 22 and 23). It was reduced and enclosed in the nineteenth century (Figure 24). Visitors can still see the original entrance, to which a gate was added when the park was encircled (Figure 25). In the middle of the park (its back today), Valentim put what he called the Fonte dos Amores (Fountain of Loves), flanked by two pyramids (Figures 26 and 27). The fountain is known as the Fonte dos Jacarés (Caiman Fountain) because of the two caimans on one side (Figure 28). These caimans are Valentim's most famous sculptures. Valentim's two bronze sculptures in the park (the lizard and the caimans) are the only works by the master still in their original location.[7] Two giant cranes from the Passeio, and statues of Echo and Narcissus as a hunter from the Marrecas fountain, can be seen today in the Rio de Janeiro Botanical Garden (Figures 29 and 30). Valentim's religious art is decidedly baroque, while his bronze sculptures are markedly neoclassical. Both, however, are idealized renderings of the sacred, in the religious art, and of nature, in the secular. No other artist contributed more than Valentim to late colonial Rio's sacred and public spaces.

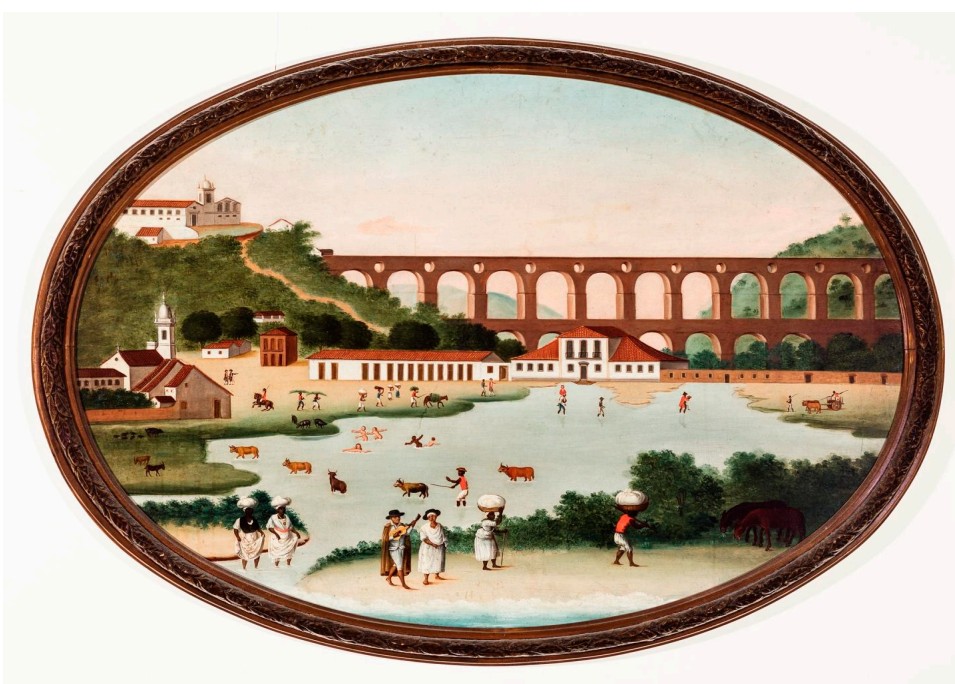

**Figure 21.** Attributed to Leandro Joaquim, *Lagoa do Boqueirão e Aqueduto da Carioca*, c. 1780. Courtesy of the Museu Histórico Nacional.

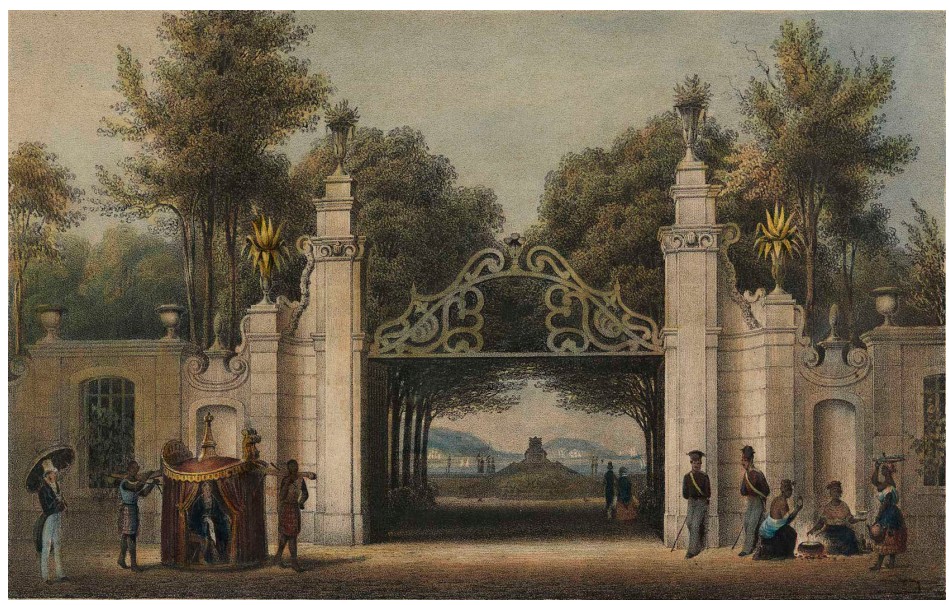

**Figure 22.** Karl Wilhelm von Theremin, *Entrance of the Passeio Público*, 1818. Courtesy of the Biblioteca Nacional.

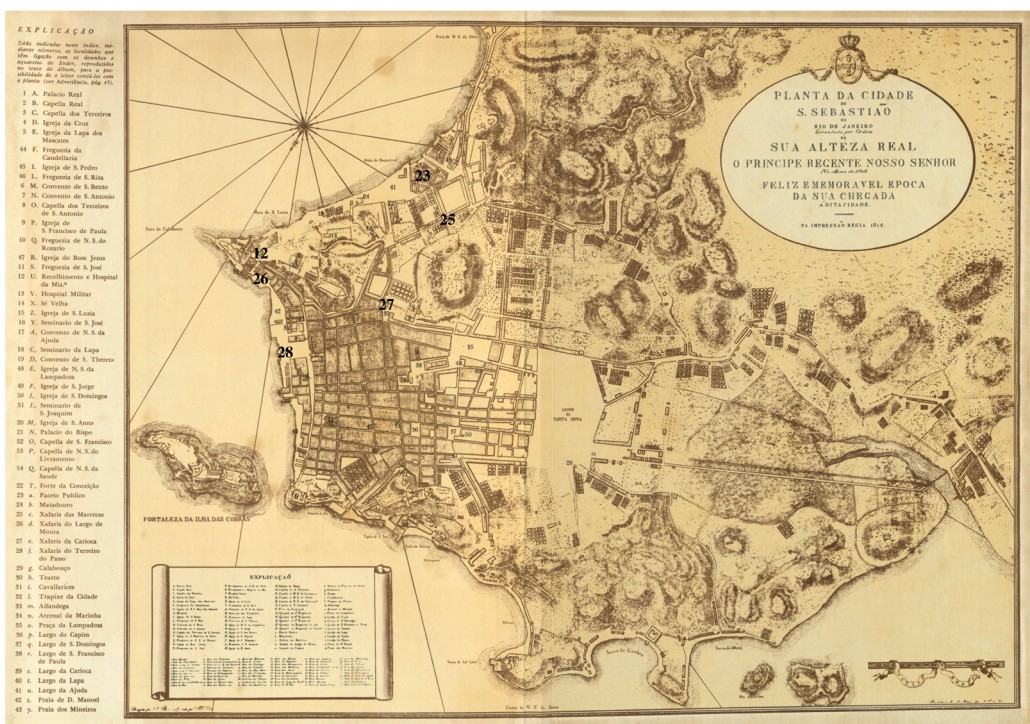

**Figure 23.** Urban plan of Rio de Janeiro in 1812 showing the location of some of Valentim's works. **12.** Recolhimento. **23.** Passeio Público. **25.** Marrecas fountain. **26.** Saracuras fountain. **27.** Lizard fountain. **28.** Pyramid fountain.

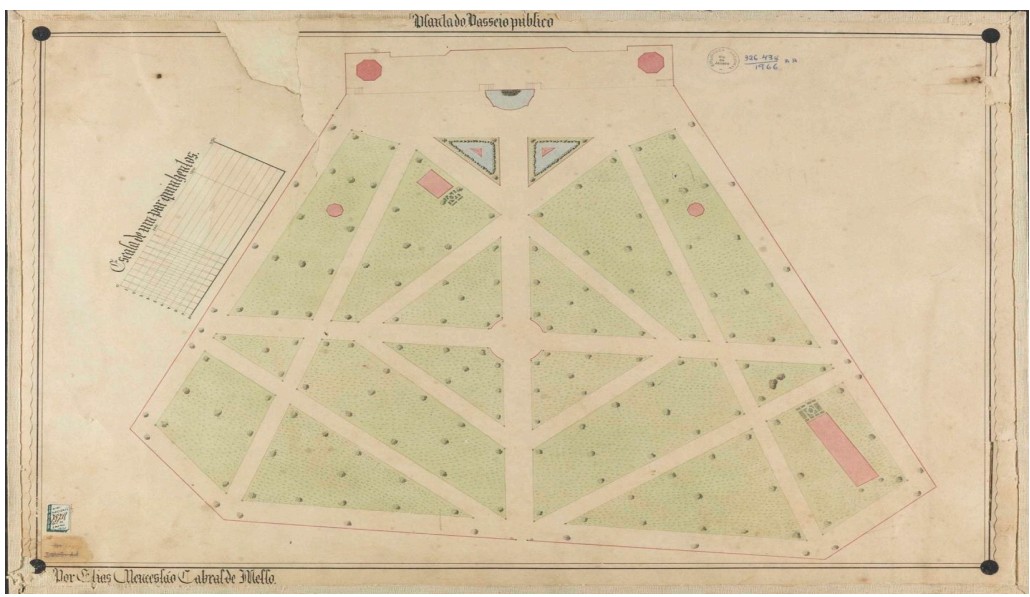

**Figure 24.** Nineteenth-century enclosure of the Passeio Público. Courtesy of the Biblioteca Nacional.

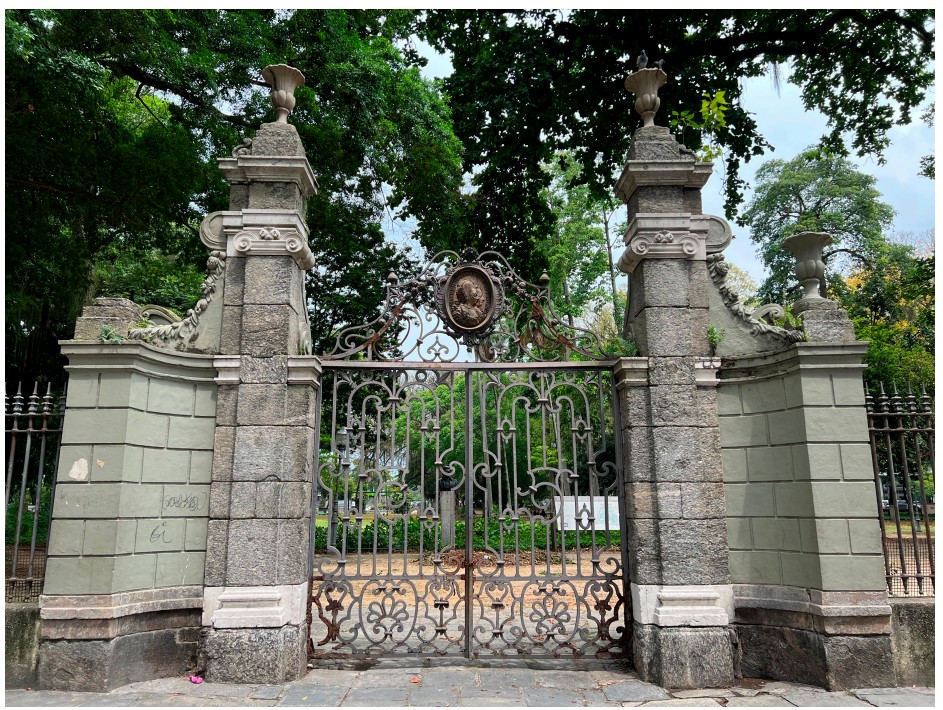

**Figure 25.** Gate of the Passeio Público today. Photo by the author, January 2023.

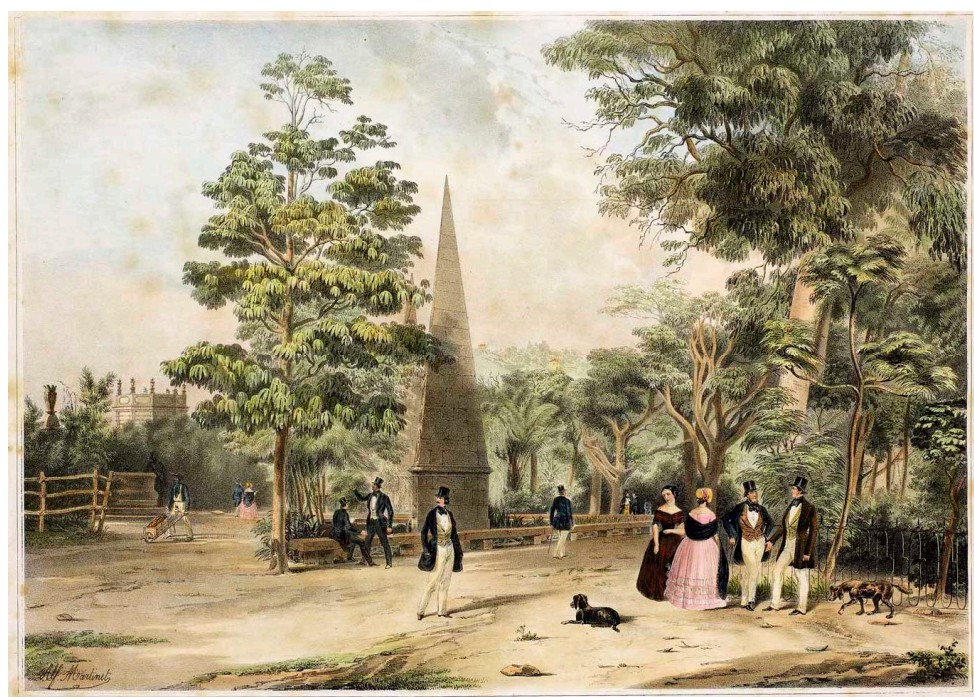

**Figure 26.** Alfred Martinet, *Passeio Público*, c. 1850. Courtesy of the Biblioteca Nacional.

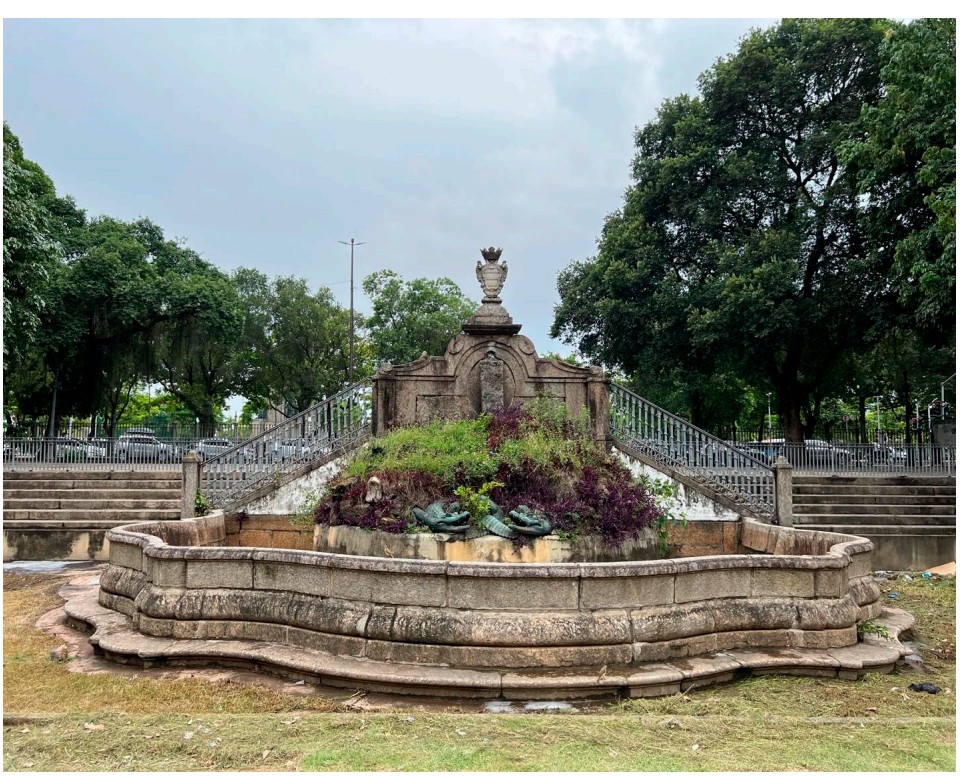

**Figure 27.** Valentim da Fonseca e Silva, Fountain of Loves, c. 1782. Photo by the author, January 2023.

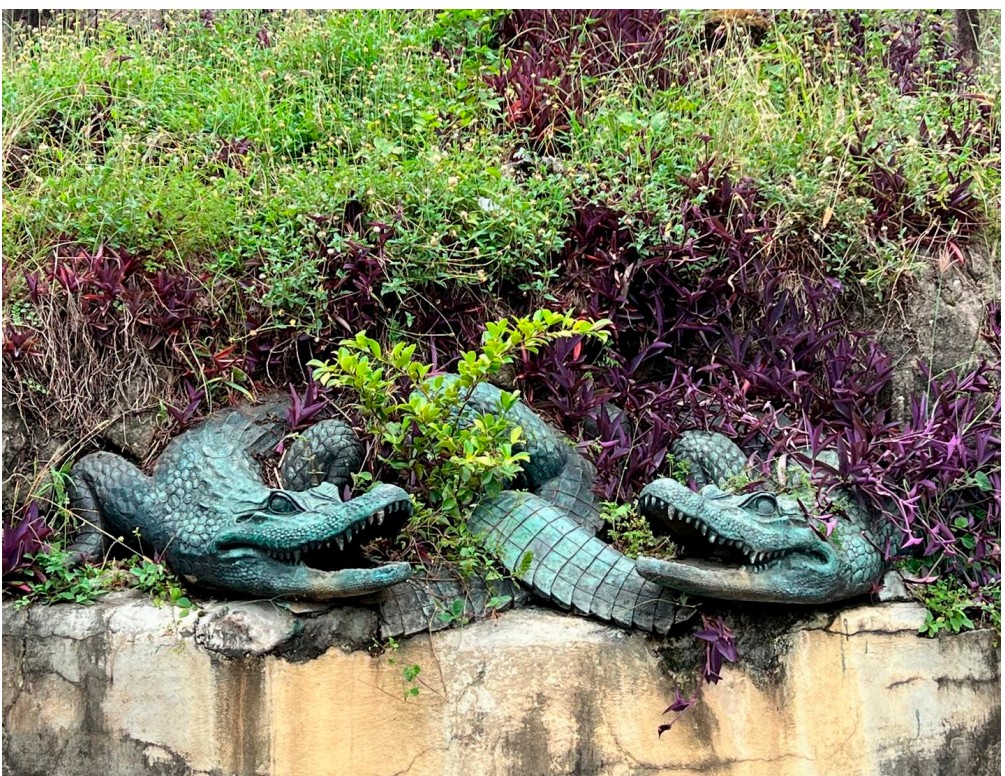

**Figure 28.** Valentim da Fonseca e Silva, Caimans from the Fountain of Loves. Photo by the author, January 2023.

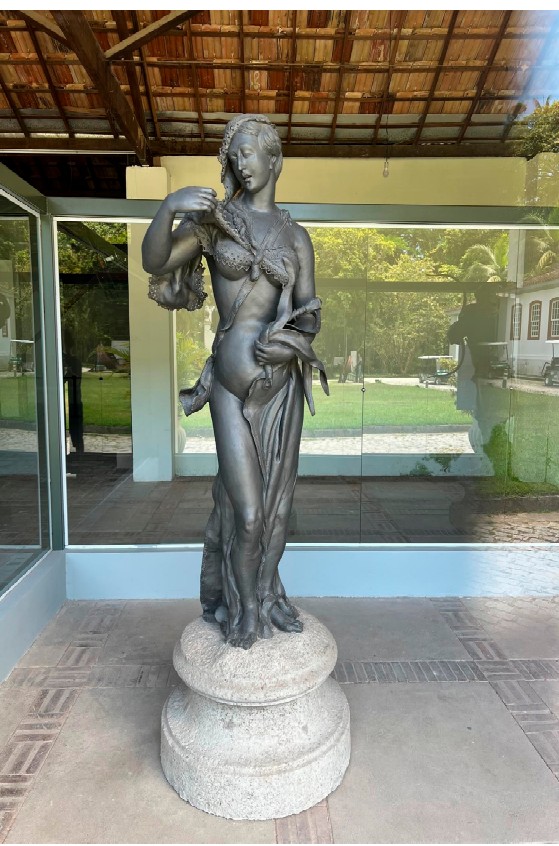

**Figure 29.** Valentim da Fonseca e Silva, Echo from Marrecas fountain, Jardim Botânico do Rio de Janeiro. Photo by the author, January 2023.

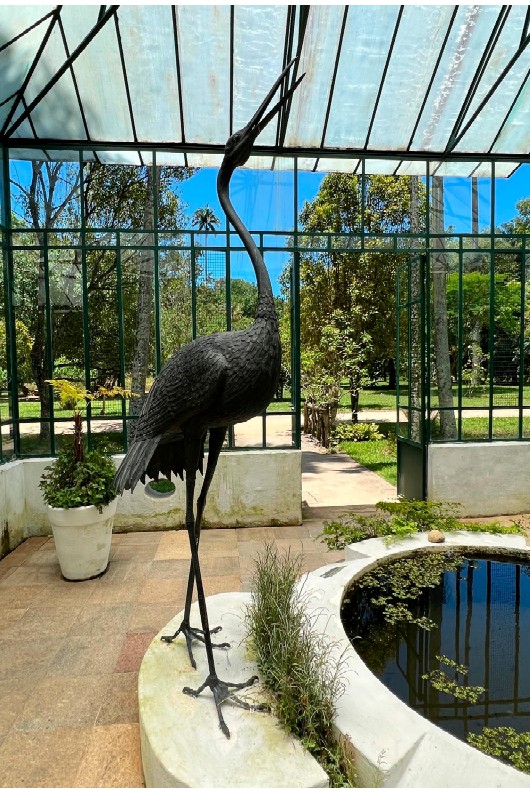

**Figure 30.** Valentim da Fonseca e Silva, Crane from Fountain of Loves, Jardim Botânico do Rio de Janeiro. Photo by the author, January 2023.

## 4. Jesus, Master Painter

Jesus, of whom we have no portrait, is the least studied of the three artists examined in this article. Like the others, very little biographical information about him is known. His date of birth, for example, is not known with exactitude. At the time of his death in poverty in 1847, his age was unknown, though he was thought to be in his nineties. If a legend about Jesus is true, then this may have been how he wanted it. According to this legend, the artist did not seek fame. One example illustrating his discretion is that, when the Brazilian Emperor Pedro I (r. 1822–31) visited Salvador in 1826, Jesus was not interested in meeting the sovereign (Campos 2003, vol. 1, p. 48). He also did not train many disciples (Campos 2003, vol. 1, p. 48).

The only trustworthy biographical document about Jesus is his marriage record in the church of the Blessed Sacrament in 1808. The document gives his parents' names as Antonio Feliciano Borges and Josefa de Santana. His mother is labeled a *crioula*, or Brazilian-born slave, and the bride is identified as a *forra* (freedwoman) born in the Gulf of Guinea (in Leite [1988] 2010, p. 42). Jesus is labeled as a pardo, a term widely used for and preferred by many mixed-race Afro-Brazilians (Valerio 2021b; Pessoa 2013; Viana 2007).

Jesus was a disciple of José Joaquim da Rocha (c. 1737–1807), who was commissioned with some of the most important artworks in Salvador, including the ceilings of Conceição da Praia (Immaculate of Conception on the Beach, one of the oldest temples in the city and an important site of Black Catholicism), the church of the Third Order of Saint Dominic (across from the Jesuit church, today's cathedral), and Our Lady of the Rosary of the Blacks in Pelourinho, the most famous Black church in Brazil (see Valerio 2021a). Rocha trained in Italy and Portugal and was a master of illusionistic paintings.[8] In the 1780s, Rocha helped Jesus travel to Europe to study with some of the most important artists of the moment (Campos 2003, vol. 1, p. 47; Querino [1911] 2018, pp. 96–102). Jesus thus belonged to a generation of Bahian pardo artists, such as Manoel José de Sousa Coutinho (mid-18th–early 18th century; dates unknown), Antônio Joaquim Franco Velasco (1780–1833, his most successful pupil), and José Verissimo de Freitas (mid-18th–early 18th century; dates unknown), formed by Rocha. While these other disciples transitioned to Neoclassism, except for his most famous secular work, Jesus "stayed closer to his baroque formation" (Campos 2003, vol. 1, p. 31). Yet, Carlos Ott, a pioneering scholar of Bahian art who reluctantly wrote about Jesus in 1980, toward the end of his career, considered Jesus "a good canvas painter, but a bad plaster painter" (Ott 1982, p. 75). Fortunately, Jesus has since been reconsidered by scholars like Maria de Fátima Hanaque Campos (2003). This revaluation of Jesus' work has vindicated what one of the first persons to write about Jesus, the Afro-Bahian polymath Manuel Querino, wrote about the artist in the early twentieth century (Querino [1911] 2018, pp. 96–102).

Returning to Salvador in the 1790s, Jesus began to receive commissions. He thus contributed to the iconography of some of Salvador's most important churches, such as the old cathedral (demolished in 1933[9]), Senhor do Bonfim (Lord of Favorable or Desired Outcome, another important site of Black Catholicism[10]), Nossa Senhora do Pilar (Our Lady of the Pillar), Third Franciscan Order, Third Carmelite Order, and the São Joaquim (St. Joachim) orphanage. He did an important work further afield, in the church of the Divina Pastora (Divine Shepherdess) in Sergipe, in Bahia's northeast Atlantic corner at the time.[11] For the eucharistic chapel of the old cathedral, he painted the *Institution of the Eucharist* (Figure 31), which is now in the Museum of Sacred Art of Bahia. In it, Jesus has reconfigured the classic distribution of the Last Supper.[12] Instead of Jesus seated surrounded by the disciples, in the foreground, a standing Jesus gives the eucharist to one of the disciples while another kisses his feet. Another disciple waits for his turn as Catholics do at the eucharist. In the background, two disciples raise their arms in exaltation while the others sit eyes closed after receiving the eucharist as Catholics are encouraged to do after communion. In this painting, then, Jesus has synthesized the Counter-Reformation Catholic doctrine on the eucharist, on which the Catholic Church placed the greatest emphasis after Trent (see Rubin 2004).

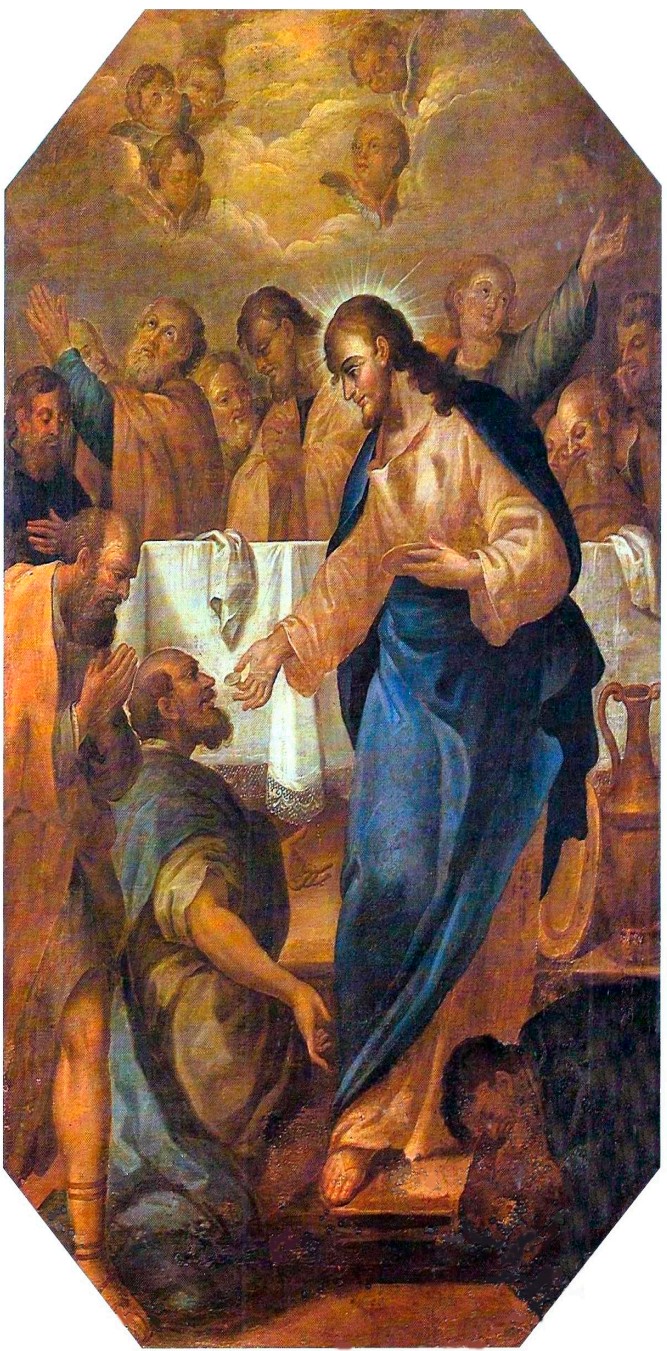

**Figure 31.** José Teófilo de Jesus, *Institution of the Eucharist*, late 18th century. Courtesy of the Museum of Sacred Art of Bahia.

In the church of the Divina Pastora in Sergipe (at the time, the northern Atlantic corner of the Captaincy of Bahia), Jesus' ceiling shows his affinity with Rocha (Figure 32). An illusionistic painting with the characteristic quadratura, or painted architectural elements, the ceiling's theme is Marian, like many of Rocha's own. At the center of the painting, we see the Virgin Mary seated in a field surrounded by sheep with Jesus on her lap. In the quadratura are scenes from the Virgin's life. At each end, painted domed tholobates let in light. Thus, compared to Rocha's darker ceilings (Figure 33), where no such painted light apparatuses exist, Jesus' composition is more luminous.

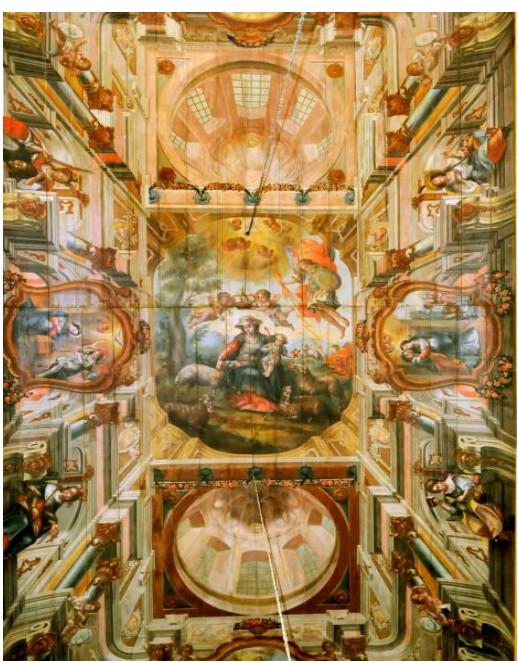

**Figure 32.** José Teófilo de Jesus, *Divina Pastora* ceiling, late 18th century. Courtesy of Wiki Commons.

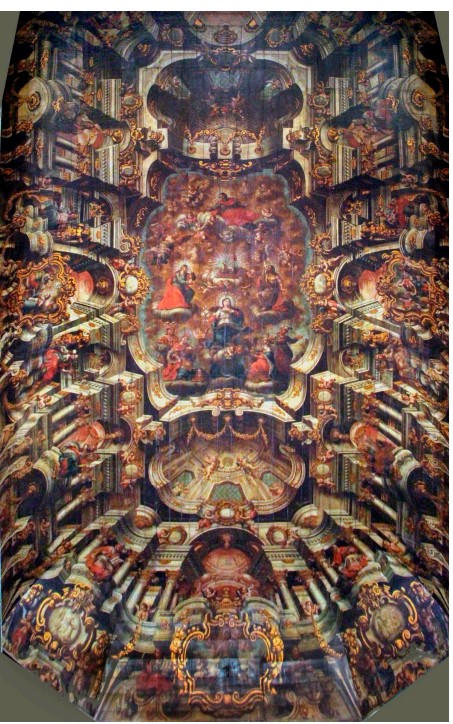

**Figure 33.** José Joaquim Rocha, *Conceição da Praia* ceiling, late 18th century. Courtesy of Wiki Commons.

Around 1809, Jesus showed his only neoclassical inclination in one of his few secular works, a set of allegories of the four continents. Jesus' series seems to have been inspired by Adriaen Collaert's (Figure 34), but with significant innovations. Like Collaert, Jesus positions the allegorical figure in an autochthonous landscape astride a representative beast: a camel for Asia, a horse for Europe, and an elephant for Africa (Figure 35), while America significantly sits on a blank cube. Collaert sat the figures on the wildest animals imaginable to him: America on an armadillo, Africa on a crocodile. Jesus followed what had become standard for Europe and Asia: the horse and the camel, respectively. America

had been represented astride different animals, like Collaert's armadillo or Johann Justin Preissler's puma. Africa was represented either astride a lion or an elephant, and sometimes a crocodile.

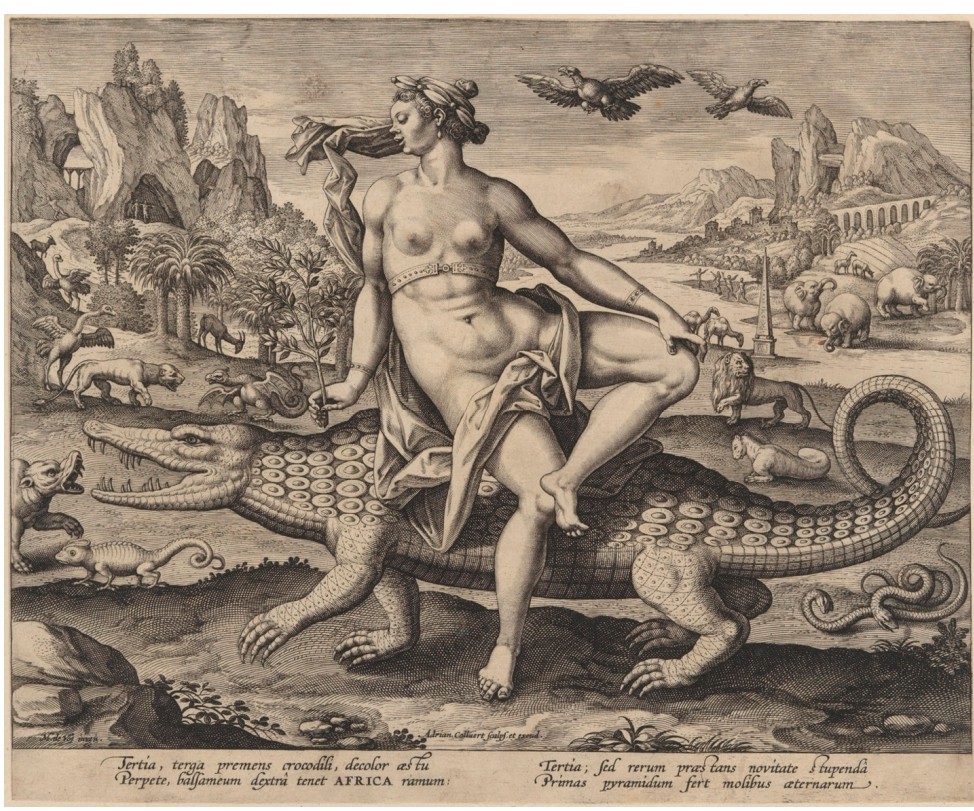

**Figure 34.** Adriaen Collaert, *Africa*, 1580–1600. Courtesy of the Metropolitan Museum of Art.

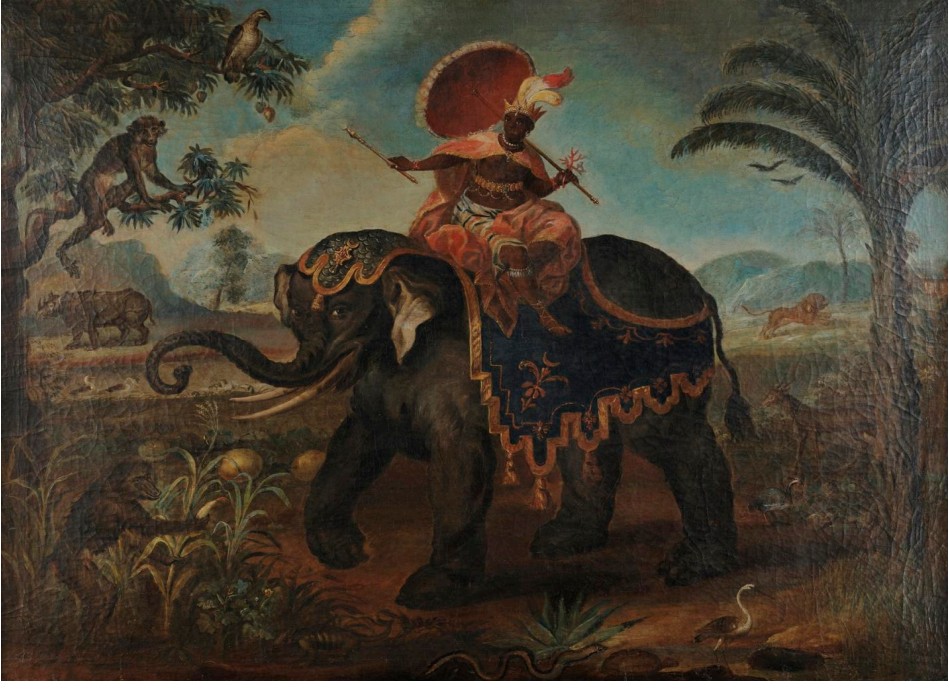

**Figure 35.** José Teófilo de Jesus, *Africa*, c. 1809. Courtesy of Museum of Art of Bahia.

In addition to the elephant replacing a dangerous-looking crocodile, a striking difference between Collaert's print and Jesus' painting is that the African figure is mostly clothed and is definitively regal. Despite the bare breasts, Jesus' painting does not present the African figure as sexually available like Collaert does. As was common in allegories of the continents, Collaert presents Europe with royal regalia while placing the other continents, Africa and the Americas in particular, at a lower hierarchical rank, most often signaled by a lack of clothing. Jesus replaces a sexualized, naked Africa surrounded by animals that signal wildness and danger with a regal figure. This completely disrupts the hierarchy presented in the other four continents' images.[13] Here, Jesus' has tamed the African elephant. In other words, he has reinscribed Africa under the banner of civilization.

Jesus' choice of the elephant for Africa may have reflected a long tradition of a performance of the four continents in which Africa appeared as a savage king riding an elephant (see Horowitz and Arizzoli 2021). While never recorded in Brazil, this tradition was well documented in Europe and elsewhere in the Americas (Valerio 2022, pp. 138–47). For example, Africa was represented this way for Philip II of Spain (r. 1546–95) in Benavente, Zamora, Spain, in 1554:

> The first performance to enter was that of a powerful elephant, very natural and beautifully made. The head of this elephant was the size of a horse with its neck and forelegs, and the other half a horse at its rump; so natural, that it was a marvel to behold. On top there was an African, only wearing a shirt, with his right sleeve rolled up, and an arrow in his hand, imitating in his posture and attire, the Indians of West Africa [*imitando en la postura y traje á los indios de las partes de África del Océano*]. (Muñoz [1554] 1877, p. 45)

The "ethnographic interchangeability" ("Indians of West Africa") by the author Andrés de Muñoz (sixteenth century; dates unknown) reflects the exotic genre within which Europeans inscribed this performance and visual allegories of Africa, America, and Asia (see Mason 1998).

A group of Afro-Mexicans staged a similar performance in 1610:

> After seeing [the first] performance, the procession came right away to another, also by Blacks, recently brought to this Kingdom from Africa, who brought an elephant of marvelous size and beauty. It was a wonder to see its figure and form so naturally done. All this machinery was set upon wheels that moved with great ease. Up above this animal there was a Black king, with a scepter in one hand and a crown on his head, convincingly representing the king of Africa. (Pérez de Rivas [1610] 1896, vol. 1, pp. 250–51)

Finally, this is how Africa appeared in Daniel Rabel's performance of the four continents in his *Le Grand Bal de la Douairière de Billebahau* in Paris in 1626 (Figure 36). Jesus may have been inspired by this tradition, which he may have witnessed while in Europe. Moreover, these performances may have been inspired by West African embassies (Lowe 2007; Ndiaye 2020), of which there were a few brought to Bahia during Jesus' lifetime, which in turn shaped local Black performance (Araujo 2012; Lara 2002; Voigt 2019). This hypothesis is strengthened by the fact that Jesus' elephant looks festive, as if it were dancing. But more than these embassies, Jesus' painting reflects the traditions brought to Bahia by the hundreds of thousands of Africans disembarked in Bahia during that time.

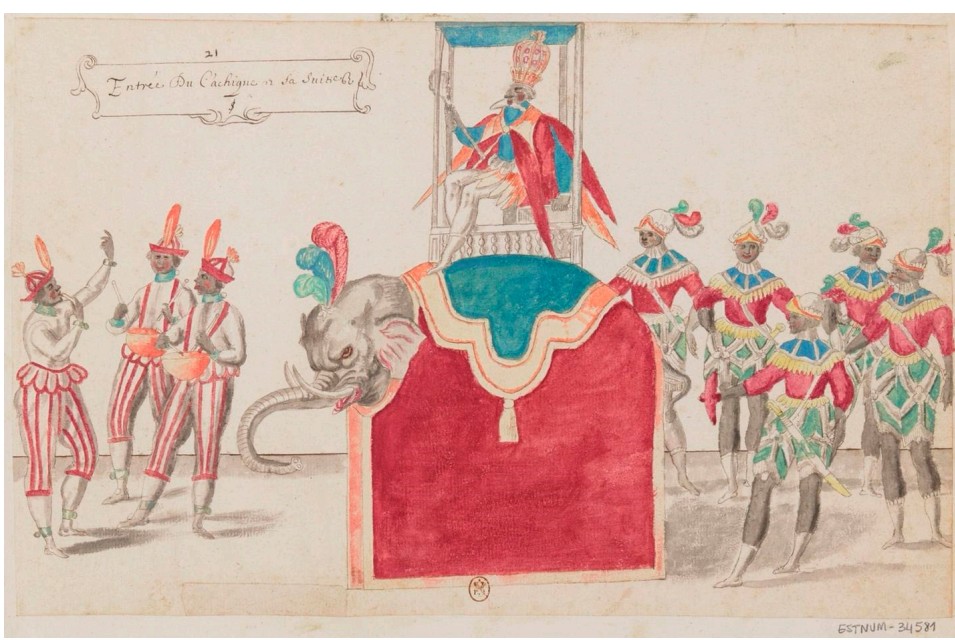

**Figure 36.** Daniel Rabel, Black "Cacique" on Elephant, for *Le Grand Bal de la Douairière de Billebahaut*, Paris, 1626. Courtesy of the Bibliothèque Nationale de France.

## 5. Conclusions

To reiterate, Aleijadinho, Valentim, and Jesus were some of the most important artists of late colonial Brazil, where they had the greatest impact on the artistic developments in their home regions. Their careers demonstrate that Afrodescendants could be important artists that defined urban and sacred spaces in the slavery-era Atlantic. We must therefore reconsider how we have viewed Afrodescendants in early modern Atlantic art. We must recognize that Afrodescendants were not only enslaved subjects that appear in the sketches and paintings of European travelers (such as the Dutch (1630–54) painter Albert Eckhout (ca. 1610–1665 (see, e.g., Parker Brienen 2006)), the Italo-Portuguese soldier Carlos Julião (1740–1811 (see, e.g., Fromont 2013)), or the French diplomat Jean-Baptiste Debret (1768–1848) (see, e.g., Wood 2013)), but that they were also artists who were held in high esteem and who transformed religious spaces and urban landscapes. As Black artists begin to receive greater institutional and scholarly attention (e.g., Pullins and Valdés 2023; Spinozzi 2022), we need to remember that many of colonial Latin America's most celebrated artists (such as Aleijadinho in Brazil and Correa in Mexico) were of African descent.

**Funding:** This research received no external funding.

**Data Availability Statement:** Not applicable.

**Acknowledgments:** I would like to thank Rachel Zimmerman for commenting on a draft of this article and for sharing myriad secondary sources with me, as well as Ana Lucia Araujo, the anonymous readers, and the academic editor for their insightful comments and suggestions. Research for this article in Brazil was made possible by grants from Washington University in St. Louis's Office of the Provost, Center for the Humanities, and Center for the Study of Race, Ethnicity, and Equity.

**Conflicts of Interest:** The author declares no conflict of interest.

## Notes

[1] On Afro-Brazilians in music, see, for example, Alves da Silva (2012), Reily (2013), Querino (Querino [1911] 2018, pp. 217–302), and Valerio (2021a).

2    On other Afro-Brazilian artists, see Araújo (Araújo [1988] 2010, vol. 1, pp. 25–182), Furtado (2015), Campos (2003), (Querino ([1911] 2018), and Rarey (2015). Other examples include José de Ibarra (Mexico City, 1685–1756; active Mexico City) (see Mues Orts 2001) and the Afro-Puerto Rican José Campeche (San Juan, 1751–1809; active San Juan) (see Thames 2022).

3    In her chronology, Anna Maria Fausto Monteiro de Carvalho (A.M.F.M. de Carvalho [1999] 2003, pp. 107–9) states that Valentim's father took him to Portugal at a young age, but this cannot be verified (see Cavalcanti 2004, pp. 310–13; da Fonseca e Silva [1813] 2008, pp. 142–44).

4    I was not able to ascertain where Valentim's remains are today.

5    The viceroy also employed the pardo painter and architect Leandro Joaquim (Rio, c. 1738–98).

6    Prior to the fire, the ex-slave and former sex worker Rosa Egipcíaca (1719–71) had ran the rocolhimento (Mott 1993; Maranhão 1997).

7    A sculpture of Cupid from the fountain has been lost.

8    The painter should not be confused with a mapmaker of the same name that lived around the same time.

9    The cathedral was demolished due to its state of advanced disrepair.

10    The Afro-Catholic ceremony of the *lavagem* ("cleansing"), which dates back to the colonial period, begins with a mass at Conceição da Praia and ends at Bonfim.

11    Ott (1982) attributes 124 works to Jesus (102–4). Ott's list does not include Jesus' allegories of the four continents.

12    Jesus painted a standard Last Supper for the Bonfim church (Ott 1982, p. 107).

13    I would like to thank the anonymous reader who specifically helped me work through this insight.

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
