# Peer review of "Atlantic Masters: Three Early Modern Afro-Brazilian Artists"

_arts, 2023_

Round 1

Reviewer 1 Report

Summary

“Masters of their World: Three Mixed-Race Colonial Afro-Brazilian Artists” is a well-written and impeccably researched study of three major eighteenth-century (and early nineteenth century) Black artists in Brazil: Aleijadinho, Mestre Valentim, and José Teófilo de Jesus. The author uses these three artists as case studies to demonstrate that, even in the Americas’ largest slavery society, Black artists dominated and mastered the colony’s arts ecosystem. As the author shows, these artists were prolific, sought-after, and acclaimed during their lifetimes. At the same time, the author argues that their level of their artistic output is potentially connected to their racial category/identification: pardo, a term broadly referring to persons with Black ancestry but of mixed-race parentage and typically possessing much lighter skin tones than those the artists enslaved to carry out some of their commissions. Thus, the author argues that these three artist’s successes must also be contextualized in, and used as an example of, the systemic colorism that structured colonial Brazilian society and its dependence on a slavery economy.

General Comments: Praise

There is much to commend in this essay. First, the case studies are well-chosen on multiple levels, providing key points of contrast that help the essay – despite its focus – provide a wide overview of the topic. The three have widely differing fame in and out of Brazil, with Aleijadinho being very well-known, while work on Teófilo de Jesus is lacking. The three also worked in distinct regions: Rio de Janeiro, Minas Gerais, and Bahia, which allows the author to show some of the key economic and cultural differences in each of these three areas. And the three had distinct artistic outputs and are known for their work in different media.

Second, though these artists are all well known to scholars of Afro-Brazilian and Brazilian Baroque art history broadly, their bibliography remains small in English and, in many cases, Portuguese. Though the essay does not carve radically new ground in interpreting the artist's works, it does help to fill an historiographic lacuna in specific ways. The author’s citations and discussion show an intimate knowledge of the Brazilian historiography and scholarship on these artists, which is often lacking in the very few English-language publications that exist. The author also has buttressed that knowledge with high-resolution photos that, to my knowledge, will be unprecedented resources in the historiography of these artists. And the author goes some way to contextualizing their work inside discourses of race and colorism in Brazil, though I believe this can be strengthened with some editing (see below).

Finally, the author very smartly avoids the trap – as is common on much recent work on these three artists – of attributing hidden African religious or syncretic meanings to their iconography as a byproduct of their racial identity; instead, he places the work in context of other Catholic and rococo / Brazilian Baroque works appropriate to the milieu in which they moved.

In general, the essay will be welcomed by specialists in the field as a contribution, as well as by undergraduate and graduate students as a resource for future work.

General Comments: Suggestions for Improvement

First, given that the essay partly seeks to “consider how colorism shaped the arts in colonial Brazil,” the essay could be improved by adding a few sentences for each artist regarding – if known – how the term pardo was operationalized by the artists during their lives. Pardo is, indeed, a still-popular term for “mixed-race” Afro-Brazilians, as the author notes (2), but this should be clarified to indicate it’s often a synonym for mulato (so, someone with white and Black ancestry; not “mixed-race” generally regardless of parentage). That said, it is also my understanding that the term is quite imprecise: as one example, Soares (People of Faith, 13) notes that in eighteenth century Rio de Janeiro parish records pardo was “somewhere between” the “extremes” of “white” and “blacks” (and both terms had connotations of blood cleanliness and social status), but it was also “difficult to grasp” what the term meant specifically. Clarifying or explaining this nuance will help on two levels: first: almost never have the racial categories around their artistic production been interrogated or contextualized comparatively (as one example, Tribe 1996, doesn’t really interrogate her use of the term mulato to describe Mestre Valentim). At the same time, the author notes – and could likely further foreground – how the artists’ status as free, combined with their enslaving of other Black artists, facilitated their artistic output and quality. Even a few additional paragraphs dedicated to the following questions would, if space allows, clarify the author’s contribution and fully distinguish it from the extant literature: How did these artists deploy the term pardo, and where does it appear? What evidence do we have of the racial categories of the artists they enslaved (or, is slave status always rendered as “Black”)? And how do the answers to each show the specific ways colorism operated in their professional lives?

Second, one could connect the above points about pardo and colorism more generally to the portraits the author reproduces. It is striking that Aleijadinho (Figure 1) is depicted with almost yellowish skin and slicked-back black hair. In contrast, Mestre Valentim is depicted as quite dark-skinned in Muzzi’s painting of 1789 (Figure 14). Perhaps this is because of the implicit contrast (and note the bowing/lowered position) of Valentim with the white officials in the painting; and Aleijadinho’s portrait is singular. But it might be worth adding just a few sentences further of analysis on some of these images to help contextualize a bit of what we’re seeing; otherwise the reader may understand these as documentary sources and I don’t think that’s the author’s intent.

Third, and connected to the above point, throughout the essay the author could add a few sentences – or even just one – describing the action or iconography in some of the images the author reproduces. This will help the reader contextualize and follow the author’s argument in visual terms, which will be especially important in an art history-themed journal.

Fourth, the essay would also be improved if the author provided some further explanation and elaboration of the wider social contexts in the distinct regions where the artists worked - Minas Gerais, Rio de Janeiro, and Bahia – at the beginning of each section. This is done quickly in the beginning, but returning to it would help readers unfamiliar with Brazilian history and increase the essay’s utility for undergraduate students in particular.

Specific Comments:

Page 2, line 43: The author refers to the “Golden Age” of Brazil ushered in by the Minas Gerais gold rush in the late seventeenth and early eighteenth centuries; but it is likely important to acknowledge that this led to a massive increase in the enslaved population as well, which gets back to the author’s argument. Thus “golden age” may not be the term the author wants here.

Page 2, line 57: The author should clarify that mineiro refers to Minas Gerais.

Page 3, line 103: The Siebers (2010) reference should be updated, as some art historians have begun to deal with disability as a framework since the publication of Disability Aesthetics thirteen years ago. It would be most welcome to see their names cited and worked acknowledged in this venue. See, for example:

·       Jessica A. Cooley, and Ann M. Fox (eds). “Indisposable: Disability and American Art.” Special Issue, Panorama: Journal of the Association of Historians of American Art, Colloquium, 8, no. 1 (Spring 2022).

·       Keri Watson and Timothy W. Hiles (eds.), Routledge Companion to Art and Disability (Routledge 2022).

·       Ann-Millett-Gallant and Elizabeth Howie (eds.), Disability and Art History (Routledge, 2017)

Page 17, line 216: The Muzzi painting is housed at the Museu da Chácara do Céu, not the Museu de Açude. These two institutions are, of course, the combined holdings of the Museus Castro Maya in Rio de Janeiro, and it may be this latter name the author needs to credit as the holding institution.

Page 17, line 221: Since most readers will not be familiar with the location, the author should clarify that Valongo is a wharf and another section of the city of Rio de Janeiro north of Praça XV.

Page 25, line 303: Given that most readers of Arts will not be familiar with Brazilian churches, I suggest the author give the full name of the Bonfim church (in this and other instances in the text). In general, the author switches between providing local Portuguese and English-translation names for churches, but these should probably be made consistent in one language or the other.

Page 25, note 15: Salvador’s pelourinho was technically the pole for all public whipping punishments, not only those for the enslaved.

Page 29: If the author believes that Teófilo de Jesus’s work is inspired by Collaert, they could do more to note the key changes he makes. I’d love to see an accounting, for example, of the sumptuous finery Teófilo de Jesus uses in Africa, in contrast to the somewhat submissive nakedness of the former. Plus, those clothes seem clearly in dialogue with Congo King celebrations (on both sides of the Atlantic - i.e. the umbrella and the cape), as the author likely knows and Fromont discusses extensively. This may be a better analogy for Teófilo de Jesus’s inspiration and helps draw from multiple sources beyond Collaert and Rabel.

Page 31, line 378: The author writes that “Pardos . . . sought to set themselves apart from Black Afro-Brazilians,” implying that pardos are not Black (and this distinction is made in the archival documents). But in the intro, the author implies that the pardo artists under discussion are both Black and Afro-Brazilian, but also enslaved other Black artists. The author should go back through the essay to be sure they are deploying the terms consistently. This also points to my above comment regarding pardo, in which it may be deployed here to describe labor and social status as much as racial identity (my own hunch is that it is used discursively to distinguish from “Black” and thus enslaved as much as it speaks to parentage, but that may not be the author’s argument).

Author Response

Thank you for your attentive reading of the article. I corrected the line by line corrections. Thank you. Page 17, line 216: I think i read that somewhere, but since i can't find I deleted the nore. I looked at the recommended art and disability lit and now engage with it further. 

Author Response

Thank you for the attentive reading of the article. I corrected the line by line corrections. Thank you. I think the revised article is stronger. Thank you. 

Reviewer 3 Report

I think a revaluation of Antonio Francisco Lisboa, Valentim Fonseca e Silva and José Teofilo de Jesus is timely and important. That said, there is an enormous literature on Afro-Brazilian studies that the author must engage, from Roberto Conduru to Lorraine Leu in the arts/urban space to general theories of Denise Ferreira da Silva to mention only a few.

Author Response

Thank you for the attentive reading of the article. I consulted the recommended authors but found little application at this stage. I think the revised article is stronger. Thank you. 

Round 2

Reviewer 3 Report

This is a much improved version but you have not yet engaged the literature on Afro-Brazilian studies: Beatriz Nascimento, Denise Ferreira da Silva, João Vargas, Christen Smith, Lorraine Leu, Marcelo Paixão, Edward Telles, Paulina Alberto, among so many others.

Author Response

I don't understand how or to what effect you want me to engage with these authors. 

In Phase 2, reader 3 recommended I engage with the works of “Beatriz Nascimento, Denise Ferreira da Silva, João Vargas, Christen Smith, Lorraine Leu, Marcelo Paixão, Edward Telles, Paulina Alberto, among so many others.” Like the authors referred in Phase 1, all these authors: 1) do not study my period or the artists I discuss in my essay; and 2) have a vast production, making it impossible for me to identify which ideas of theirs I should engage, although I read as much as possible of theirs. From I what I consulted, I did not find anything that I could engage with in my essay, since these authors center on very localized aspects of twentieth-century Brazil. In my essay’s current form, I have abandoned a racialized approach, and focus instead on the special issue theme of Black artists in the Atlantic. I therefore find this reader’s suggestions inapplicable.